# Carbonaceous dust grains seen in the first billion years of cosmic time

Joris Witstok[1,2 ✉], Irene Shivaei[3,25 ✉], Renske Smit[4,25 ✉], Roberto Maiolino[1,2,5], Stefano Carniani[6], Emma Curtis-Lake[7], Pierre Ferruit[8], Santiago Arribas[9], Andrew J. Bunker[10], Alex J. Cameron[10], Stephane Charlot[11], Jacopo Chevallard[10], Mirko Curti[1,2,12], Anna de Graaff[13], Francesco D'Eugenio[1,2], Giovanna Giardino[14], Tobias J. Looser[1,2], Tim Rawle[15], Bruno Rodríguez del Pino[9], Chris Willott[16], Stacey Alberts[3], William M. Baker[1,2], Kristan Boyett[17,18], Eiichi Egami[3], Daniel J. Eisenstein[19], Ryan Endsley[20], Kevin N. Hainline[3], Zhiyuan Ji[3], Benjamin D. Johnson[19], Nimisha Kumari[21], Jianwei Lyu[3], Erica Nelson[22], Michele Perna[9], Marcia Rieke[3], Brant E. Robertson[23], Lester Sandles[1,2], Aayush Saxena[5,10], Jan Scholtz[1,2], Fengwu Sun[3], Sandro Tacchella[1,2], Christina C. Williams[24] & Christopher N. A. Willmer[3]

Large dust reservoirs (up to approximately $10^8 M_\odot$) have been detected[1–3] in galaxies out to redshift $z \simeq 8$, when the age of the Universe was only about 600 Myr. Generating substantial amounts of dust within such a short timescale has proven challenging for theories of dust formation[4,5] and has prompted the revision of the modelling of potential sites of dust production[6–8], such as the atmospheres of asymptotic giant branch stars in low-metallicity environments, supernova ejecta and the accelerated growth of grains in the interstellar medium. However, degeneracies between different evolutionary pathways remain when the total dust mass of galaxies is the only available observable. Here we report observations of the 2,175 Å dust attenuation feature, which is well known in the Milky Way and galaxies at $z \lesssim 3$ (refs. 9–11), in the near-infrared spectra of galaxies up to $z \simeq 7$, corresponding to the first billion years of cosmic time. The relatively short timescale implied for the formation of carbonaceous grains giving rise to this feature[12] suggests a rapid production process, possibly in Wolf–Rayet stars or supernova ejecta.

As part of the James Webb Space Telescope (JWST) Advanced Deep Extragalactic Survey (JADES), we obtained deep Near-Infrared Spectrograph (NIRSpec) multi-object spectroscopic data taken in the PRISM configuration (spectral range 0.6 to 5.3 μm and resolving power $R \simeq 100$). Using the NIRSpec micro-shutter array (MSA), we observed 253 sources across three visits between 21 and 25 October 2022 (JWST programme 1210; principal investigator (PI): Lützgendorf), with exposure times per object ranging from 9.3 to 28 h. The extracted one-dimensional spectra reached a continuum sensitivity ($3\sigma$) of approximately $6$–$40 \times 10^{-22}$ erg s$^{-1}$ cm$^{-2}$ Å$^{-1}$ (27.2–29.1 AB magnitude) at approximately 2 μm. Targets were selected with a specific focus on high-redshift galaxies in imaging taken with the Hubble Space Telescope and JWST/Near-Infrared Camera (NIRCam).

Through a visual inspection of all spectra, we found strong evidence of an absorption feature around a rest-frame wavelength $\lambda_{emit} = 2,175$ Å

in the spectrum of a galaxy at $z = 6.71$ (JADES-GS+53.15138−27.81917; JADES-GS-z6-0 hereafter), which was revealed via a significant ($6\sigma$) deviation from a smooth power-law continuum, as shown in Fig. 1. This feature, known as the ultraviolet (UV) attenuation 'bump', was first discovered by Stecher (1965) along sightlines in the Milky Way[9] and is attributed to carbonaceous dust grains, specifically polycyclic aromatic hydrocarbons (PAHs) or nano-sized graphitic grains[12]. We fitted a Drude profile around 2,175 Å to the excess attenuation[13], defined as the observed spectrum normalized to a bump-free attenuated spectrum that is predicted by a power-law function fitted outside the bump region (Methods). We found a bump strength (amplitude) of $0.43^{+0.07}_{-0.07}$ mag and a central wavelength $\lambda_{max} = 2,263^{+20}_{-24}$ Å. The latter, although within the range expected by models of carbonaceous grains[14], is notably higher than the range typically observed along different sightlines in the Milky Way, potentially suggestive of a change in grain mixture[15]. Beyond the local

[1]Kavli Institute for Cosmology, University of Cambridge, Cambridge, UK. [2]Cavendish Laboratory, University of Cambridge, Cambridge, UK. [3]Steward Observatory, University of Arizona, Tucson, AZ, USA. [4]Astrophysics Research Institute, Liverpool John Moores University, Liverpool, UK. [5]Department of Physics and Astronomy, University College London, London, UK. [6]Scuola Normale Superiore, Pisa, Italy. [7]Centre for Astrophysics Research, Department of Physics, Astronomy and Mathematics, University of Hertfordshire, Hatfield, UK. [8]European Space Agency, European Space Astronomy Centre, Madrid, Spain. [9]Centro de Astrobiología, CSIC–INTA, Madrid, Spain. [10]Department of Physics, University of Oxford, Oxford, UK. [11]Sorbonne Université, CNRS, Institut d'Astrophysique de Paris, Paris, France. [12]European Southern Observatory, Garching bei München, Germany. [13]Max-Planck-Institut für Astronomie, Heidelberg, Germany. [14]ATG Europe for the European Space Agency, ESTEC, Noordwijk, the Netherlands. [15]European Space Agency, Space Telescope Science Institute, Baltimore, MD, USA. [16]NRC Herzberg, Victoria, British Columbia, Canada. [17]School of Physics, University of Melbourne, Parkville, Victoria, Australia. [18]ARC Centre of Excellence for All Sky Astrophysics in 3 Dimensions (ASTRO 3D), https://astro3d.org.au/. [19]Center for Astrophysics | Harvard & Smithsonian, Cambridge, MA, USA. [20]Department of Astronomy, University of Texas, Austin, TX, USA. [21]AURA for European Space Agency, Space Telescope Science Institute, Baltimore, MD, USA. [22]Department for Astrophysical and Planetary Science, University of Colorado, Boulder, CO, USA. [23]Department of Astronomy and Astrophysics, University of California, Santa Cruz, Santa Cruz, CA, USA. [24]NSF's National Optical-Infrared Astronomy Research Laboratory, Tucson, AZ, USA. [25]These authors contributed equally: Irene Shivaei, Renske Smit. ✉e-mail: jnw30@cam.ac.uk; ishivaei@cab.inta-csic.es; r.smit@ljmu.ac.uk

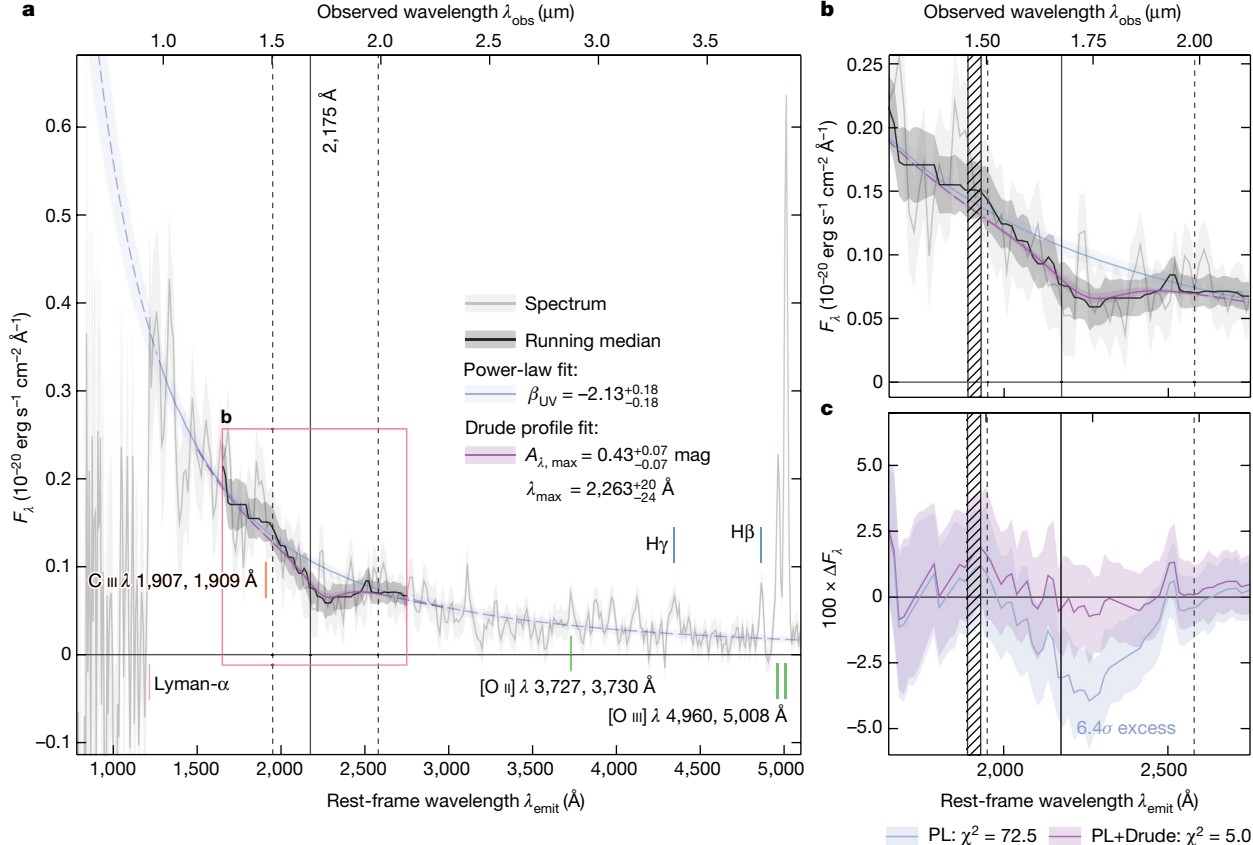

**Fig. 1 | Spectrum taken by JWST/NIRSpec of JADES-GS-z6-0 at redshift**
**z = 6.71. a**, Overview of the spectrum (grey solid line) with a power-law fit to the
UV continuum (blue solid line). Several spectral features used to confirm the
spectroscopic redshift are indicated, including the Lyman-α break, the [O II] λ
3,727, 3,730 Å doublet, and the Hβ, Hγ and [O III] λ 4,960, 5,008 Å lines.
**b**, Zoom-in of the UV bump region around $\lambda_{emit} = 2,175$ Å, where a running median
(solid black line), representing the attenuated stellar continuum, reveals a deep
localized absorption profile. A Drude profile fit within the vertical dashed lines

(purple solid line) with respect to the smooth power law (blue solid line) yields
an amplitude of $0.43^{+0.07}_{-0.07}$ mag and a central wavelength $\lambda_{max} = 2,263^{+20}_{-24}$ Å. The
hatched region indicates the C III λ 1,907, 1,909 Å doublet. **c**, The residuals ($\Delta F_\lambda$)
show that the power-law (PL) fit alone has a significant negative flux excess
between approximately 2,000 and 2,400 Å (6.4σ), whereas the power-law fit
and Drude profile combined (PL+Drude; purple line) provides a significantly
better fit ($\chi^2 = 72.5$ versus $\chi^2 = 5.0$ for PL and PL+Drude, respectively). All
shading represents 1σ uncertainty.

Universe, the feature has previously been observed spectroscopically
only in massive, metal-enriched galaxies at $z \lesssim 3$, suggesting it origi-
nates in dust grains exclusively present in evolved galaxies[11,13,16–19].
The detection reported here is a direct, spectroscopic detection of
the UV bump in a galaxy at $z > 3$.

The properties of JADES-GS-z6-0 are summarized in Table 1. This
galaxy shows significant dust obscuration as probed by the ratio of
Balmer lines (the 'Balmer decrement'). Here, Hα/Hβ ≃ 3.7 indicates
a nebular extinction of $E(B − V)_{neb} = 0.25 ± 0.07$ mag. In agreement
with the trend between metallicity and bump strength observed at
lower redshift, measurements of the gas-phase and stellar metallicities
($Z \simeq 0.2–0.3 Z_\odot$) further suggest that JADES-GS-z6-0 has undergone
substantial metal enrichment relative to galaxies with a similar mass
at the same redshift[20].

To systematically investigate the prevalence of the UV bump and
obtain clues about its origin at such early times, we selected JADES
galaxies with a confident spectroscopic redshift above $z > 4$ with a
median signal-to-noise ratio of at least 3 per spectral pixel in the region
corresponding to rest-frame wavelengths of $1,268$ Å $< \lambda_{emit} < 2,580$ Å.
This resulted in a sample of 49 objects between redshift 4.02 and 11.48.
Comparing the continuum slopes on both sides of the central wave-
length at 2,175 Å (Methods), we selected ten galaxies (at $4.02 < z < 7.20$)
from this parent sample whose spectral shape indicates the presence
of a UV bump.

We constructed a weighted average (or 'stack'; Methods) of all 49
objects in our parent sample as well as a stack for the ten objects with
evidence for a bump signature, as shown in Fig. 2. In both stacks, we
find evidence for emission from the C III λ 1,907, 1,909 Å nebular lines
that are commonly seen in metal-poor galaxies[21]. There is no indication
of the bump in the parent sample. The stacked spectrum of the ten
selected objects, however, shows a clear depression (5σ) centred on
approximately 2,175 Å. Although we did not find evidence for significant
differences in stellar properties (mass or age), these ten galaxies are
characterized by a considerable amount of dust obscuration, compa-
rable to $z \simeq 2$ galaxies with the bump feature[18], and mildly enhanced
metallicities compared to the parent sample (Methods). We again fit-
ted a Drude profile to the excess attenuation in the stacked spectrum
of these ten objects, finding a bump amplitude of $0.10^{+0.01}_{-0.01}$ mag and a
central wavelength $\lambda_{max} = 2,236^{+21}_{-20}$ Å.

Although the UV bump has long been known to exist, its variable
presence and strength have been an open topic of debate in galaxy
evolution studies[13,22]. The feature is commonly attributed to PAHs[12],
molecules thought to be susceptible to destruction by hard ionizing
radiation, and it is present in the Milky Way and Large Magel-
lanic Cloud extinction curves but is very weak or absent in the Small Magel-
lanic Cloud curves[23]. In the attenuation curve of individual galaxies,
radiative-transfer effects determined by the dust-star geometry can
weaken the bump in the observed integrated spectrum[24,25]. However,

## Table 1 | Properties of JADES-GS-z6-0

| Property | Value |
|---|---|
| RA (deg) | +53.15138 |
| Dec (deg) | −27.81917 |
| $t_{exp}$ (h) | 27.9 |
| $z_{spec}$ | $6.70647^{+0.00044}_{-0.00033}$ |
| $m_{F115W}$ (mag) | 28.58 ± 0.12 |
| $M_{UV}$ (mag) | −18.34 ± 0.12 |
| $\beta_{UV}$ | $-2.13^{+0.18}_{-0.18}$ |
| $\gamma_{34}$ | $-3.6^{+1.5}_{-1.2}$ |
| $Z_{neb}$ ($Z_\odot$) | $0.17^{+0.05}_{-0.04}$ |
| $E(B-V)_{neb}$ (mag) | 0.25 ± 0.07 |
| $M_*$ ($10^8 M_\odot$) | $1.0^{+0.3}_{-0.2}$ |
| $Z_*$ ($Z_\odot$) | $0.34^{+0.05}_{-0.05}$ |
| $SFR_{30}$ ($M_\odot\,yr^{-1}$) | $3.0^{+2.0}_{-1.0}$ |
| $t_*$ (Myr) | $18^{+11}_{-7}$ |

Error bars represent 1σ uncertainty. Rows: (1) right ascension (RA) in J2000; (2) declination (Dec) in J2000; (3) exposure time ($t_{exp}$) in the NIRSpec PRISM spectra in hours; (4) spectroscopic redshift ($z_{spec}$); (5) apparent AB magnitude in the NIRCam F115W filter ($m_{F115W}$); (6) absolute AB magnitude in the UV ($M_{UV}$); (7) UV spectral slope ($\beta_{UV}$); (8) spectral slope change around $\lambda_{emit} = 2,175$ Å ($\gamma_{34}$); (9) gas-phase metallicity ($Z_{neb}$; from rest-frame optical emission lines) in units of solar metallicity; (10) nebular extinction ($E(B-V)_{neb}$; from the Balmer decrement) in magnitudes; (11) stellar mass ($M_*$) in $10^8$ solar masses; (12) stellar metallicity ($Z_*$; from SED modelling) in units of solar metallicity; (13) SFR in solar masses per year averaged on a timescale of 30 Myr ($SFR_{30}$); (14) mass-weighted stellar age ($t_*$) in Myr.

by stacking the photometry of large samples of galaxies, the bump has been detected to varying degrees at redshifts $z \lesssim 3$, with tentative hints at $z \lesssim 6$ (refs. 11,17,19,26). Spectroscopically, the bump has been seen only in relatively massive and dusty individual galaxies at $z \simeq 2$ (refs. 13,18). In Fig. 3, the bump amplitude is shown as a function of cosmic time, including its strength in the extinction curves of Milky Way, Large Magellanic Cloud and Small Magellanic Cloud sightlines[23]. Our inferred bump amplitude and central wavelength, especially in the individual spectrum of JADES-GS-z6-0, are comparatively high, the former defying the trend with stellar mass seen at lower redshift. This may point towards a different nature of the grains responsible for the absorption (for example, graphite instead of PAHs) in addition to a different, possibly simpler, dust–star geometry compared with lower-redshift counterparts. Intriguingly, there is tentative evidence for a colour gradient in JADES-GS-z6-0 (Methods).

Moreover, a direct detection of the bump at $z \simeq 4$–7 is striking given that at these redshifts, the age of the Universe is only around a billion years (approximately 800 Myr at $z = 6.71$). Substantial production of carbon and the subsequent formation of carbonaceous grains responsible for the absorption feature through the standard asymptotic giant branch (AGB) channel, particularly in the low-metallicity regime characterizing such early galaxies ($Z \simeq 0.1 Z_\odot$; ref. 20), would require low-mass ($M \lesssim 2.5 M_\odot$) and, hence, long-lived stars to reach the AGB at the end of their lives, after more than 300 Myr (ref. 7). If this is the dominant channel via which carbonaceous grains are formed, the presence of the UV bump implies the onset of star formation in these galaxies occurred within the first half billion years of cosmic time, corresponding to redshift $z \gtrsim 10$. Indeed, star formation has been shown to occur at this early epoch with the confirmation of $z > 10$ galaxies[27]. However, in our sample, we did not find evidence for substantial star formation activity that occurred on timescales beyond 300 Myr (Methods). The absence of clear signatures from such relatively old stellar populations suggests that other, faster channels for the production of carbonaceous dust are required in these early systems, corroborated

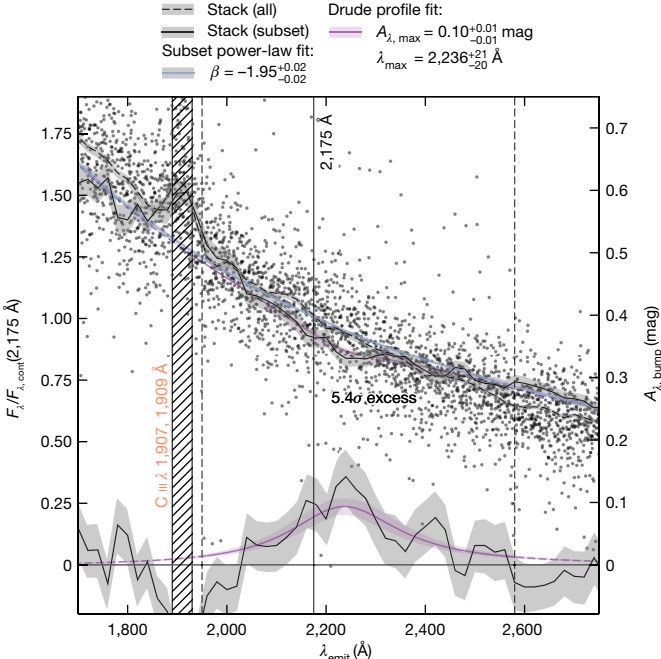

**Fig. 2 | Normalized and stacked spectra around the UV bump of $z > 4$ JADES galaxies observed by JWST/NIRSpec.** Spectra of all galaxies (small black dots) are shifted to the rest frame and normalized to the predicted continuum level at a rest-frame wavelength of $\lambda_{emit} = 2,175$ Å in the absence of a UV bump (Methods). The dashed black line (shading represents 1σ uncertainty) is a stacked spectrum obtained by combining all 49 objects in wavelength bins of $\Delta\lambda_{emit} = 20$ Å. The hatched region clearly shows emission from the C III $\lambda$1,907, 1,909 Å doublet. The stacked spectrum of ten galaxies selected to have a bump signature (solid black line, shading as 1σ uncertainty), in addition to appearing to have a mildly redder UV slope, shows the presence of the UV bump around 2,175 Å. This bump has an excess with respect to a power-law continuum (solid blue line; see Methods) at a significance of 5.4σ. The excess attenuation $A_{\lambda,bump}$ (curve at the bottom, corresponding to the axis on the right) is fitted with a Drude profile (shown in purple with shading as 1σ uncertainty), which gives an amplitude of $0.10^{+0.01}_{-0.01}$ mag and a central wavelength $\lambda_{max} = 2,236^{+21}_{-20}$ Å.

by the high observed frequency of extremely metal-poor Milky Way stars that are carbon enhanced[28].

One explanation is that these grains formed on considerably shorter timescales via more massive and rapidly evolving stars, possibly by supernovae (SNe) or Wolf–Rayet (WR) stars, which would overhaul some, and place strong constraints on other, theoretical models of dust production and stellar evolution. PAH production has, indeed, been observed in WR stars[29], and while subsequent SN type-Ib/c explosions are generally expected to destroy most dust produced in the preceding WR phase, models have shown that carbonaceous grains produced by binary carbon-rich WR stars can survive[30]. However, for standard initial mass functions, WR stars, especially carbon-rich WR stars, are rare[31]. Conversely, isotopic signatures in presolar graphite grains found on primitive meteorites indicate a type-II SN origin, suggesting the production of these potential carriers of the UV bump starts at early times[32]. Indeed, dust production in SN ejecta has been regarded as a potential rapid channel for significant dust production in the early Universe[33], its net efficiency depending on the grain destruction rate in the subsequent reverse shock[34]. However, substantial carbonaceous production in SN ejecta is expected only by some classes of models and for a certain subclass of scenarios (for example, non-rotating progenitors), whereas other models favour the formation of silicates or other types of dust[35–38]. In summary, our detection of carbonaceous dust at $z \simeq 4$–7 provides crucial constraints on the dust production models and scenarios in the early Universe.

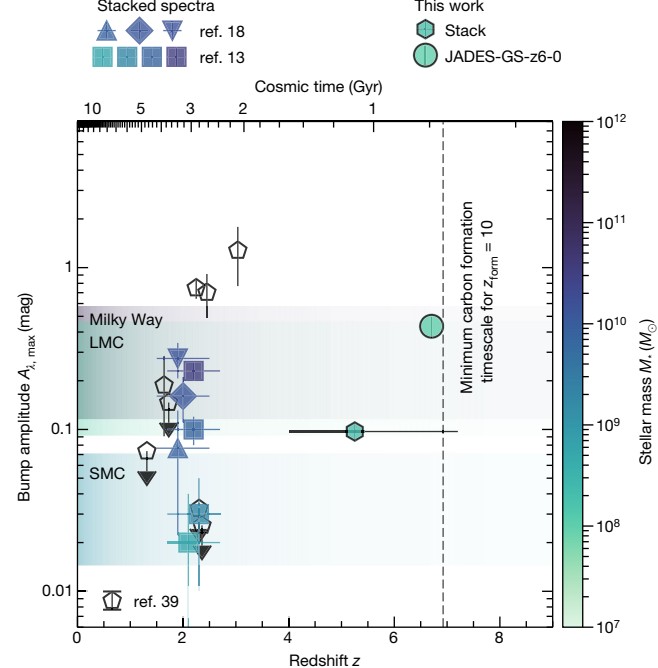

**Fig. 3 | Redshift evolution of UV bump strength.** Amplitude of the excess attenuation $A_{\lambda,\mathrm{max}}$ is shown for JADES-GS-z6-0 individually as well as for the stack of ten $z \simeq 4$–$7$ JADES galaxies. Points are coloured according to their (average) stellar mass. Error bars along the $y$ axis represent $1\sigma$ uncertainty. At $z \simeq 2$, measurements from γ-ray burst absorbers (ref. 39 and references therein) and from stacked spectra in various bins of stellar mass[13] or shape of the UV continuum as a whole and in the bump region[18] are shown (see Methods for details)[13,18,39]. Error bars of the stacked spectra along the $x$ axis represent the full redshift range, their central values slightly shifted for visualization purposes. The bump amplitudes in the average Milky Way (MW), Large Magellanic Cloud (LMC) and Small Magellanic Cloud (SMC) dust extinction curves[23,40], converted to an attenuation for a visual extinction range of 0.1 mag $< A_V <$ 0.5 mag, are indicated with light shading. The age of the Universe is indicated at the top. The vertical dashed line indicates the minimum timescale required for carbon production by AGB stars (that is, 300 Myr) if the galaxy formed at $z_{\mathrm{form}} = 10$.

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

## Methods

### Data and parent sample

The observations presented here were taken as part of JADES[41], a joint survey conducted by the JWST[42,43] NIRCam[44] and NIRSpec[45,46] Guaranteed Time Observations instrument science teams. As described in Robertson et al.[47] and Curtis-Lake et al.[27], deep NIRCam imaging[48] over a wavelength range $\lambda_{obs} \simeq 0.8$ to 5 μm (reaching $m_{AB} \simeq 30$ mag in F200W) was taken under JWST programme 1180 (PI: Eisenstein) in an area of 65 arcmin$^2$ over the Great Observatories Origins Deep Survey−South (GOODS-S), which includes the Hubble Ultra Deep Field. We additionally made use of public medium-band imaging taken as part of the JWST Extragalactic Medium-band Survey (JEMS[49]; JWST programme 1963, PI: Williams) and First Reionization Epoch Spectroscopic Complete Survey (FRESCO[50]; JWST programme 1895, PI: Oesch). By incorporating the wealth of publicly available ancillary data from the Hubble Space Telescope, a catalogue with photometric redshifts was constructed and used to identify high-redshift galaxy candidates. NIRSpec multi-object spectroscopy[51] of these NIRCam-selected sources was performed with the MSA[52] in the PRISM/CLEAR spectral configuration, covering a spectral range 0.6 to 5.3 μm with resolving power $R \simeq 100$. A three-point nodding pattern was implemented for background subtraction in addition to small dithers with MSA reconfigurations to increase sensitivity and flux accuracy, improve spatial sampling, mitigate the impact of detector gaps and aid the removal of cosmic rays. Dither pointings consisted of four sequences of three nodded exposures. Each setup was made up of two integrations of 19 groups, resulting in exposure times of 8,403.2 s for each sequence and of 33,612 s (9.3 h) for each dither pointing[27]. The main galaxy considered in this work (JADES-GS-z6-0) was observed in three visits, resulting in an integration time of 27.9 h (Table 1), whereas the other targets had exposure times ranging between 9.3 and 27.9 h. The spectral energy distribution (SED) and a false-colour image of JADES-GS-z6-0 with the location of the NIRSpec MSA shutters overlaid are shown in Extended Data Fig. 1. Note that the imaging reveals a tentative colour gradient, with the shutter capturing the redder part of the galaxy, which may contribute to the strength of the UV bump in the spectrum of JADES-GS-z6-0.

Flux-calibrated two-dimensional spectra and one-dimensional spectral extractions were obtained with pipelines developed by the European Space Agency's NIRSpec Science Operations Team and the NIRSpec Guaranteed Time Observations team, which will be discussed in detail in a forthcoming paper. The pipelines generally apply the same algorithms in the official Space Telescope Science Institute pipeline that generates Mikulski Archive for Space Telescopes products. An irregular wavelength grid with 5 spectral pixels per resolution element was adopted to avoid oversampling the line spread function at short wavelengths ($\lambda_{obs} \simeq 1$ μm). The one-dimensional spectra were extracted with a 5 pixel aperture covering the entire shutter size to recover the entire emission. However, as in Curtis-Lake et al.[27], we considered an additional extraction over a 3 pixel aperture to test the robustness of our findings, as discussed in Robustness of the UV bump detections. Given the compact sizes of the high-redshift galaxies considered here (Extended Data Fig. 1), slit-loss corrections were applied under the assumption of a point-like source placed at the relative intrashutter position of each galaxy. We note that systematic uncertainties in the slit-loss correction are a smooth function of wavelength and did not, therefore, affect the UV bump signature, which instead relies on detecting the UV slope inflection over a relatively small wavelength range around $\lambda_{emit} = 2,175$ Å (as discussed in the next sections). Extraction was performed in a shutter-size aperture to recover all emission. Further details regarding the target selection and data reduction are extensively discussed in preceding JADES works[27,41,47,48,51].

### Sample selection

After running an automated spectral fitting routine in BAGPIPES (Bayesian Analysis of Galaxies for Physical Inference and Parameter Estimation)[53], spectroscopic redshift estimates were confirmed by visual inspection independently by at least two team members. The final redshift values were determined by a subsequent analysis (described in detail in Chevallard et al., in prep.) with BEAGLE (Bayesian Analysis of Galaxy SEDs)[54], as described in Curtis-Lake et al.[27] but with a star formation history (SFH) consisting of a 10-Myr-long star formation burst combined with a delayed exponential component, and a narrow redshift prior distribution centred around the visually confirmed redshifts. We selected objects with a confident spectroscopic redshift above $z > 4$ to ensure the rest-frame UV coverage included the Lyman-α break. Based on the formal uncertainty, we further selected spectra with a median signal-to-noise ratio of at least 3 in the region corresponding to rest-frame wavelengths of $1,268$ Å $< \lambda_{emit} < 2,580$ Å.

We then performed several Bayesian power-law fitting procedures to the rest-frame UV continuum with a Python implementation[55] of the MultiNest[56] nested sampling algorithm. To identify spectra exhibiting a UV bump, we fitted power laws in four adjacent wavelength windows defined by Noll et al.[57] (with corresponding power-law indices $\gamma_1$ to $\gamma_4$), excluding the region $1,920$ Å $< \lambda_{emit} < 1,950$ Å to avoid contamination by the C III doublet. In the presence of the UV bump, the spectral shape of the rest-frame UV is characterized by a strong turnover in the power-law slope directly blue- and redwards of 2,175 Å covered by regions 3 and 4 respectively, resulting in a negative $\gamma_{34} \equiv \gamma_3 - \gamma_4$ value. Before fitting these separate wavelength windows in the individual spectrum, we applied a running median filter over 15 spectral pixels that cover three times the spectral resolution. We estimated the uncertainty on the running median with a bootstrapping procedure in which we randomly perturbed each of the 15 spectral pixels according to their formal uncertainty for 100 iterations.

In the fitting algorithm, a likelihood was calculated based on the inverse-variance weighted squared residuals between a given model and the observed spectrum within the adopted spectral regions. We chose flat prior distributions for the power-law indices (in the range $-5 < \gamma_i < 1$) and normalization at the centre of each wavelength window (between 0 and twice the maximum value of the spectrum in the fitting regions). Best-fit values of $\gamma_{34}$, whose posterior distribution was obtained from simultaneously fitting $\gamma_3$ and $\gamma_4$, are shown in Extended Data Fig. 2 as the 50th percentile (the median) with 16th and 84th percentiles as a ±1σ confidence range. A selection of galaxies with a median value of $\gamma_{34} < -1$, in addition to $\gamma_{34} < 0$ within the 1σ uncertainty range, led to the identification of ten galaxies (including JADES-GS-z6-0) with evidence for a UV bump (the 'bump sample'). Next, we discuss the physical properties of this subsample in the context of the full sample. Coordinates and other properties of these ten galaxies are reported in Extended Data Table 1.

### Physical properties

We consistently used a flat lambda cold dark matter (ΛCDM) cosmology based on the results of the Planck collaboration[58] (that is, $H_0 = 67.4$ km s$^{-1}$ Mpc$^{-1}$ and $\Omega_m = 0.315$). Several of the main physical properties of the full sample are presented in Extended Data Fig. 2. Extended Data Table 1 lists the observed properties of the ten individual galaxies in the bump sample. Extended Data Table 2 reports median values for the bump sample, the sample of galaxies not contained in the bump sample (the 'non-bump sample') and the full sample, as well as values measured from the stacked spectra.

**UV magnitudes and slopes.** We derived UV magnitudes directly from NIRCam photometric data points probing a rest-frame wavelength of approximately 1,500 Å (F115W for JADES-GS-z6-0; see Table 1), if available. Note that several targets fall outside the NIRCam footprint. We fitted an overall UV slope $\beta_{UV}$ to the rest-frame UV continuum probed by the NIRSpec PRISM measurements using a similar Bayesian power-law fitting procedure as described in Sample selection. We adopted the spectral windows defined by Calzetti et al.[59], which were

designed to exclude several UV emission and absorption features. Indeed, no strong emission lines were observed within these spectral regions of our low-resolution spectra. Importantly, they explicitly exclude the bump region and C III emission lines. We chose a Gaussian prior distribution for the power-law index (centred on $\mu_\beta = -2$ with a width of $\sigma_\beta = 0.5$) and a flat prior on the normalization at $\lambda_{emit} = 1,500$ Å (between 0 and twice the maximum value of the spectrum in the fitting regions). The resulting UV slope of JADES-GS-z6-0 is reported in Table 1.

**Spectroscopic rest-frame optical properties.** Emission line fluxes in the NIRSpec PRISM measurements of the individual galaxies in our sample were obtained using the pPXF software[60] (for details, refer to Curti et al.[20]). We converted Hα/Hβ line ratios into a nebular extinction $E(B - V)_{neb}$ with the Cardelli et al.[61] extinction curve, assuming an intrinsic ratio Hα/Hβ = 2.86 appropriate for case-B recombination, $T_e = 10^4$ K and $n_e = 100$ cm$^{-3}$ (for example, ref. 62). Note that for JADES-GS+53.13423-27.76891 at $z = 7.0493$, the Hα line is precisely on the edge the PRISM spectral coverage, causing the measured Hα/Hβ ratio to appear significantly below the theoretical value of Hα/Hβ = 2.86 expected in the absence of dust. Moreover, we caution that potential wavelength-dependent slit-loss effects could bias the Hα/Hβ measurements (although minimally, as the objects in this analysis are only marginally resolved) and that the stellar and nebular extinctions have a non-trivial dependence. However, despite such systematic uncertainties, galaxies strongly obscured by dust are still expected to be identifiable via their elevated Hα/Hβ line ratios.

The gas-phase oxygen abundances in our sample were derived primarily by exploiting the detection of multiple emission lines, where available, in NIRSpec medium-resolution ($R \simeq 1,000$) grating/filter configurations (G140M/F070LP, G235M/F170LP and G395M/F290LP) taken alongside the PRISM spectroscopic observations (details are discussed in Curti et al.[20]). For targets that were not covered by $R \simeq 1,000$ observations, the PRISM spectra were considered. More specifically, we required a minimum 3σ detection of [O III] $\lambda$ 5,008 Å, [O II] $\lambda$ 3,727, 3,730 Å, [Ne III] $\lambda$ 3,870 Å and Hβ before we included these lines in the metallicity calculation. For the detected emission lines, we combined information from the R3, R23, O32 and Ne3O2 line-ratio diagnostics, adopting the calibrations described in Nakajima et al.[63]. When only [O III] $\lambda$ 5,008 Å and Hβ were detected, and therefore, R3 was the only available line ratio, upper limits on [O II] $\lambda$ 3,727, 3,730 Å and [N II] $\lambda$ 6,584 Å were exploited to discriminate between the high- and low-metallicity solutions of the double-branched R3 calibration. The full procedure is described in more detail in Curti et al.[20]. We quote the gas-phase metallicity ($Z_{neb}$) in units of solar metallicity ($Z_\odot$), assuming 12 + (O/H)$_\odot$ = 8.69 as the solar oxygen abundance[64].

We further explored the rest-frame optical properties of our samples by considering composite spectra around the strong optical emission lines in Extended Data Fig. 3. These stacked spectra were obtained equivalently as described in Spectral stacking, but with bins of $\Delta\lambda_{emit} = 10$ Å given the increased spectral resolution of NIRSpec at longer wavelengths[45]. To study the Balmer decrement, we included only galaxies for which Hα is observable (that is, we did not consider objects at $z > 7.1$, leaving out one source in the bump sample). We obtained fluxes of the main emission lines (that is, [O II] $\lambda$ 3,727, 3,730 Å, [O III] $\lambda$ 4,960, 5,008 Å, Hβ and Hα) by fitting Gaussian profiles, as shown in Extended Data Fig. 3. The measured line ratios are reported in Extended Data Table 2.

**Stellar population synthesis modelling.** We employed the BAGPIPES code[53] to model the SED, which is simultaneously probed by NIRSpec PRISM measurements and NIRCam photometry, for which we used a conservative 10% error floor. For the underlying stellar models, we used the Binary Population and Spectral Synthesis (BPASS[31]) v2.2.1 stellar population synthesis models, which include binary stars.

We used the default BPASS initial mass function with a slope of −2.35 (for $M > 0.5 M_\odot$) and a range of stellar masses from 1 $M_\odot$ to 300 $M_\odot$. Aiming for a model that is simple yet able to capture older stellar populations, we adopted a constant SFH with a minimum age varying between 0 (that is, ongoing star formation) and 500 Myr, and a maximum age varying between 1 Myr and the age of the Universe. The total stellar mass formed varied between 0 and $10^{15} M_\odot$ and the stellar metallicity between 0 and 1.5 $Z_\odot$. Nebular emission was included in a self-consistent manner using a grid of Cloudy[65] models parametrized by the ionization parameter ($-3 < \log_{10} U < -0.5$). We chose a flexible Charlot and Fall[66] dust attenuation prescription with varying visual extinction ($0 < A_V < 7$ mag) and power-law slope ($0.4 < n < 1.5$). We fixed the fraction of attenuation arising from stellar birth clouds to 60% (the remaining fraction originating in the diffuse interstellar medium; for example, see ref. 67). Note that the Calzetti et al.[59] dust attenuation curve yielded consistent results. A first-order Chebyshev polynomial (described in Carnall et al.[68]) was included to account for aperture and flux-calibration effects in the spectroscopic data. The detailed properties of JADES-GS-z6-0 are reported in Table 1. Moreover, the resulting stellar masses ($M_*$), star formation rates (SFRs) averaged over the last 30 Myr (SFR$_{30}$), and mass-weighted stellar ages ($t_*$) inferred from SED models of the entire sample are presented in Extended Data Fig. 2. Median values of all properties for the galaxy sample with and without evidence for a UV bump are reported in Extended Data Table 2.

**Stellar population age determination.** Further, we explored whether the apparent absence of a significantly older stellar population ($t_* > 300$ Myr) could be explained by an 'outshining effect' due to a more recent burst of star formation[69]. Indeed, there is evidence that a substantial fraction (20% to 25%) of reionization-era galaxies ($z \gtrsim 6$) host such evolved stellar populations[70,71]. Taking the best-fitting parameter values in our BAGPIPES model, we added an instantaneous burst of star formation to the original model with a single (constant) SFH component. Comparing the reduced chi-squared values between the original, single-component model and the new, two-component model (accounting for an additional three model parameters, namely stellar mass, metallicity and age of the burst), we inferred, from a stellar population synthesis modelling point of view, how large a stellar mass can be while being 'disguised' in an evolved stellar population. This is illustrated in Extended Data Fig. 4, which shows the age-sensitive 4000 Å (Balmer) break. To avoid systematic uncertainties due to flux calibration or slit losses in the spectrum, we restricted the chi-squared analysis to the photometry. We determined the difference in reduced chi-squared values as $\Delta\chi_\nu^2 = \chi_{\nu,evolved}^2 - \chi_{\nu,original}^2$, where $\chi_{\nu,original}^2$ ($\chi_{\nu,evolved}^2$) is the reduced chi-squared metric of the single-component (two-component) model. From this conservative estimate, we cannot definitively rule out the existence of an additional population of evolved stars. For example, for $\Delta\chi_\nu^2 = 4$ (that is, at 2σ or 95% confidence), up to $5.5 \times 10^7 M_\odot$ ($9.6 \times 10^7 M_\odot$) or 0.55× (0.95×) of the inferred stellar mass of JADES-GS-z6-0 could have been produced in a 250 (500)-Myr-old burst of star formation. This scenario, however, where a galaxy builds up more than half of its stellar mass following an extended period (that is, more than 250 Myr) with little or no star formation, is physically implausible given the smooth SFH expected for relatively massive galaxies in this early epoch ($M_* \gtrsim 10^8 M_\odot$)[72]. Even a more stochastic mode of star formation is not likely to undergo such a lengthy quiescent period, suggesting that the SEDs should reveal detectable signatures of stars with intermediate ages (approximately 100 Myr), if star formation activity can truly be traced back over a time period required for AGB stars to produce substantial amounts of dust. Instead, we constrain any additional 100-Myr-old component to have at most 0.31× the current stellar mass (approximately $3 \times 10^7 M_\odot$; 2σ). This suggests that more than half, if not most, of the stellar mass in JADES-GS-z6-0 was built up in less than 100 Myr. Finally, we note that stacked rest-frame optical spectra (discussed in Spectroscopic rest-frame optical properties),

when normalized to the continuum at $\lambda_{emit} \simeq 3600$ Å, equally do not reveal a strong Balmer break either in the bump sample or in the full sample, further supporting the finding that these galaxies have relatively young stellar populations.

**Ancillary far-infrared observations.** To search for additional signatures of dust obscuration, we considered archival Atacama Large Millimeter/submillimeter Array (ALMA) 1.2 mm and 3 mm continuum imaging taken in GOODS-S. All sources in our sample were contained within the combined 1.2 mm data of the ALMA twenty-six arcmin[2] survey of GOODS-S at one millimeter (ASAGAO[73]; ALMA project code 2015.1.00098.S, PI: K. Kohno), which includes the ALMA Hubble Ultra Deep Field survey[74] (project code 2012.1.00173.S, PI: J. Dunlop) and the GOODS-ALMA survey[75] (project code 2015.1.00543.S, PI: D. Elbaz) and reaches a continuum sensitivity of approximately 78 μJy (3σ). A further 15 sources, including three sources in the bump sample (JADES-GS+53.17022-27.77739, JADES-GS+53.16743-27.77548 and JADES-GS+53.16660-27.77240), are covered by the ALMA Spectroscopic Survey (ASPECS[76,77]; project code 2013.1.00146.S, PI: F. Walter), reaching a 3σ continuum sensitivity of approximately 38 μJy at 1.2 mm and approximately 11.4 μJy at 3 mm. None of the 49 sources in our sample, however, show a significant detection (3.5σ) in either dataset. A stacking procedure, similarly, does not yield any detectable continuum emission, neither for the sources in the bump sample nor for the full sample, indicating that the non-detections can be explained by the relatively low sensitivity of the ALMA mosaics. Indeed, we have verified that for a typical SFR of a few solar masses per year (as inferred for JADES-GS-z6-0), even a conservatively high fraction (50%) of dust-obscured star formation results in an infrared luminosity that requires several tens of hours to secure a confident detection ($L_{IR} \simeq 10^{10} L_{\odot}$, translating to a continuum flux density of $F_{\nu} \simeq 5$ μJy in band 6).

**Bump parametrization and fitting procedure**
Given an observed flux density profile $F_{\lambda}$, we parametrized the UV bump profile by defining the excess attenuation as in Shivaei et al.[13]: $A_{\lambda,bump} = -2.5 \log_{10}(F_{\lambda}/F_{\lambda,cont})$. For the individual spectrum of JADES-GS-z6-0, we took the power-law fit with UV slope $\beta_{UV}$ measured outside the bump region as the attenuated spectrum without a bump, $F_{\lambda,cont}$. When considering the excess attenuation in the individual spectrum of JADES-GS-z6-0, we again used the running median and corresponding uncertainty (described in Sample selection), which was additionally used to compute the significance of the negative flux excess of the spectrum with respect to the power-law fit alone (Fig. 1c). Note that the formal uncertainty of each spectral pixel is scaled upwards to include the effects of covariance between adjacent pixels. We have verified that a similarly high significance is found when bootstrapping a spectrum first rebinned to match the spectral resolution element (thereby largely negating the effects of correlated noise). Using the MultiNest[56] nested sampling algorithm, we fitted the excess attenuation $A_{\lambda,bump}$ with a Drude profile[78], which has been shown to appropriately describe the spectral shape of the bump[13,18,79]. Centred on rest-frame wavelength $\lambda_{max}$, it is parametrized as

$$A_{\lambda,bump} = A_{\lambda,max} \frac{\gamma^2/\lambda^2}{\left(1/\lambda^2 - 1/\lambda_{max}^2\right)^2 + \gamma^2/\lambda^2}$$

where the full-width at half-maximum is $\gamma\lambda_{max}^2$. We fixed $\gamma = 250$ Å/$(2,175$ Å$)^2$ which, if $\lambda_{max} = 2,175$ Å, corresponds to a full-width at half-maximum of 250 Å, in agreement with what has been found for $z \simeq 2$ star-forming galaxies[13,18]. Again motivated by the spectral windows defined by Calzetti et al.[59], we fitted the data in a region of 1,950 Å $\leq \lambda_{emit} \leq 2,580$ Å (reflecting the $\gamma_3$ and $\gamma_4$ regions discussed in Sample selection), which excludes the C III doublet. As a prior for the bump amplitude $A_{\lambda,max}$, we conservatively chose a gamma distribution with shape parameter $a = 1$ and scale $\theta = 0.2$, which favours the

lowest amplitudes (noting that a flat prior yields comparable results). For the central wavelength, we adopted a flat prior in the range 2,100 Å $< \lambda_{max} <$ 2,300 Å.

**Spectral stacking**
For the spectral stacking analysis, we shifted each spectrum to rest-frame wavelengths $\lambda_{emit}$ and normalized it to the value of the power-law fit at $\lambda_{emit} = 2,175$ Å. The individual continuum spectra and corresponding uncertainties are rebinned to bins of $\Delta\lambda_{emit} = 20$ Å using SpectRes[80]. Stacked continuum profiles were created by weighting each binned data point by its inverse variance, although note that we obtained similar results with an unweighted average. The stacked continuum profile $F_{\lambda}$ of the ten galaxies with evidence for a UV bump was converted to an excess attenuation, as described in Bump parametrization and fitting procedure, where for the 'bumpless' profile ($F_{\lambda,cont}$), we refitted a power-law continuum to the stacked continuum profile of the ten galaxies, noting the difference in slope (measured to be $\beta \simeq -1.95$ compared to the stacked spectrum of the full sample of 49 galaxies (with $\beta \simeq -2.12$; Extended Data Table 2). To ensure good agreement with the observed continuum outside the region used in the bump fitting procedure, this power law was determined from the Calzetti et al.[59] windows bluewards of $\lambda_{emit} = 1,850$ Å (explicitly excluding the C III doublet region), whereas at wavelengths beyond the bump region, we consider the windows 2,500 Å $< \lambda_{emit} <$ 2,600 Å and 2,850 Å $< \lambda_{emit} <$ 3,000 Å (avoiding potential Mg II doublet emission at $\lambda_{emit} \simeq 2,800$ Å). Fitting a Drude profile[78] yields an amplitude of $0.10^{+0.01}_{-0.01}$ mag and a central wavelength $\lambda_{max} = 2,236^{+21}_{-20}$ Å. Note that the amplitude remains effectively unchanged if we instead fix the central wavelength to $\lambda_{max} = 2,175$ Å.

**Robustness of the UV bump detections**
**JADES-GS-z6-0.** To test the robustness of the identification of the UV bump in JADES-GS-z6-0, we extracted one-dimensional spectra from the three separate observing visits to show that the feature around 2,175 Å is not dominated by a single observation. This is illustrated in Extended Data Fig. 5, which shows measurements from each individual visit normalized to its power-law fit. We furthermore tested our extraction of the one-dimensional spectra using different apertures on the reduced two-dimensional spectra (see Data and parent sample). This slightly lowered the average continuum flux level and signal-to-noise ratio, but we found no significant changes to the rest-frame UV spectrum. We also compared NIRCam apodized photometry (the total background-subtracted NIRCam flux passing through the NIRSpec MSA slit) to synthetic photometry obtained from convolving the PRISM spectra with NIRCam filters. We verified that for most sources, the two fluxes are offset by a constant factor with offsets smoothly varying as a function of wavelength. Finally, note that the attenuation feature is a highly localized region in the low-resolution PRISM spectra (a rest-frame width of 250 Å is sampled by approximately six independent spectral resolution elements at a resolution of $R(\lambda_{obs} \simeq 1.7$ μm$) \simeq 50$) such that its magnitude is not significantly affected by the absolute flux calibration. Moreover, this wavelength range was probed by more than ten native detector pixels, indicating that the chances that this feature was produced by correlated detector noise or other artefacts are minimal.

**Stacked spectra.** In this section, we discuss the robustness of the identification of the bump feature in our stacked spectra. First, we randomly split our bump sample into two subsamples and confirmed the bump signature is present in both, implying that the stacked spectrum is not dominated by a single source. Further, we verified that performing an analogous stacking procedure at a different wavelength (2,475 Å) for a subset of sources selected based on the continuum shape around 2,475 Å in an equivalent manner as the $\gamma_{34}$ selection described in Sample selection does not produce a significant broad absorption feature as

in Fig. 2. Instead, the result was a narrow negative excess with positive excess on the edge of our fitting window, hence yielding a substantially lower amplitude when fitted with a Drude profile.

We now explore various properties of the different samples measured by NIRSpec and NIRCam to test whether the bump signature could purely be due to random noise fluctuations, in which case the ten selected galaxies are expected to simply be a random subset of the parent sample. As seen in Extended Data Fig. 2, we found a significant correlation ($p < 0.05$ for the null hypothesis that the data are uncorrelated) between on the one hand the UV slope inflection around 2,175 Å, $\gamma_{34}$, and on the other hand the absolute UV magnitude $M_{UV}$. Our selected bump sample is measured to be intrinsically fainter in the rest-frame UV (higher $M_{UV}$, independent of the SED modelling). This may be indicative of the absence of young stellar populations, in line with the theoretically predicted trend of decreasing bump strength with increasing star formation activity, and hence the intensity and hardness of the UV radiation field[81]. Moreover, several of the median properties hint at systematically different physical conditions in the galaxies part of the bump sample. In particular, these objects exhibit a significantly enhanced Hα/Hβ ratio, indicating that on average the nebular emission in these galaxies experiences a higher degree of dust obscuration, with nebular extinction values comparable to those of $z \simeq 2$ galaxies with a UV bump[18]. Moreover, their slightly elevated gas-phase oxygen abundances indicate that they are more highly enriched in metal (Extended Data Table 2). Interestingly, however, the stellar masses of the bump sample are substantially lower than their $z \simeq 2$ counterparts, as illustrated in Fig. 3. Note that other factors, such as geometry, could play an important role in determining the strength of the UV bump, although larger samples are needed to confirm these trends.

To avoid potential biases in the correlations based on individual galaxy properties due to contaminants in our $\gamma_{34}$-selected sample, we explored the stacked spectra. For instance, note that the bump and non-bump samples appear to be characterized by a comparable median UV slope, as measured in the individual spectra, which is confirmed by the agreement with the UV slopes in the unweighted stacked spectra. However, the weighted stacked spectrum shown in Fig. 2 reveals that the bump sample has a significantly redder UV continuum (as discussed in Spectral stacking). From the stacked spectra around the strong optical emission lines in Extended Data Fig. 3, we again found the Hα/Hβ ratio in the bump sample was significantly higher, translating to a nebular extinction $E(B-V)_{neb}$ a factor of approximately 2 higher than in the stacked spectrum of the full sample. This indicates that the bump sample preferably contains dustier galaxies, strongly favouring the interpretation that the observed excess attenuation around 2,175 Å is due to dust absorption. Moreover, we found evidence for a mildly higher metallicity in the bump sample through an enhanced line ratio of [O III] $\lambda$ 5,008 Å to Hβ. Although this line ratio follows a double-branched metallicity solution (for example, ref. 20), a low-metallicity solution that monotonically increases with [O III]/Hβ should be appropriate for the current sample of galaxies, given the [O III]/[O II] line ratio of approximately 10 (both in the full sample and the subset of sources selected as the bump sample). Note that such differences in the Hα/Hβ and [O III]/Hβ line ratios are absent in the control sample discussed above, which was selected based on the continuum shape around 2,475 Å.

Finally, we verified that a blind selection from the parent sample of sources with the highest Balmer decrements and reddest UV slopes resulted in a tentative detection of the UV bump. Specifically, requiring a Balmer decrement of Hα/Hβ $\gtrsim 4$ and a UV slope of $\beta_{UV} \gtrsim -2.2$ yielded a sample of four sources all contained within the bump sample (namely JADES-GS+53.16871-27.81516, JADES-GS+53.13284-27.80186, JADES-GS+53.17022-27.77739 and JADES-GS+53.16743-27.77548; Extended Data Table 1) but notably excluded JADES-GS-z6-0. Without any preselection for the continuum shape around 2,175 Å, the stacked spectrum of these four galaxies produced a tentative (approximately $4\sigma$) bump feature.

## Bump amplitude comparison with literature results

As discussed in Shivaei et al.[13], the adopted parametrization of bump amplitude in the excess attenuation (that is, $A_{\lambda,max}$; see Bump parametrization and fitting) includes the extinction in the absence of the bump $E(B-V)$ to avoid propagating the large uncertainties of this parameter that stem from the not well-constrained assumptions on the attenuation curves of high-redshift galaxies. Note that a direct measurement of the Balmer recombination line ratios can, in principle, constrain the nebular extinction[82], but its relation with stellar extinction carries uncertainty in addition to suffering from wavelength-dependent slit-loss effects (also discussed in Spectroscopic rest-frame optical properties). In Fig. 3, we directly compare these excess attenuation strengths, taking into account the underlying extinction $E(B-V)$ for bump strengths measured for the Milky Way, Large Magellanic Cloud and Small Magellanic Cloud extinction curves. In terms of the commonly used Fitzpatrick and Massa[40,78,83] parametrization, $A_{\lambda,max} = c_3/\gamma^2 E(B-V)$. We retrieve $E(B-V)$ from the measured total-to-selective extinction $R_V = A_V/E(B-V)$ for each extinction curve, assuming a range of 0.1 mag $< A_V < 0.5$ mag. Data points from Noll et al.[18] and Heintz et al.[39] (and references therein) were similarly converted to a consistent bump amplitude $A_{\lambda,max}$ using their measured values of $E(B-V)$. Measurements from Noll et al.[18] represent the stacked spectra of three subsamples that were selected based on their UV slope $\beta_{UV}$ and bump strength parametrized by the $\gamma_{34}$ parameter (Sample selection). The upwards-pointing triangle in Fig. 3 has $\beta_{UV} < -1.5$ and $\gamma_{34} > -2$, the diamond has $\beta_{UV} > -1.5$ and $\gamma_{34} > -2$, and the downwards-pointing triangle has $\gamma_{34} < -2$. Note that the Heintz et al.[39] measurements of γ-ray burst absorbers are effectively along a line of sight through the galaxies, whereas the Shivaei et al.[13] and Noll et al.[18] measurements, like the measurements in this work, are based on the total integrated light of galaxies. The distribution of dust with respect to the stars within galaxies affects the latter, integrated observations of the UV bump[24,84].

## Data availability

The data that support the findings of this study are publicly available[48,51] at https://archive.stsci.edu/hlsp/jades. Source data are provided with this paper.

## Code availability

The AstroPy[85,86] software suite is publicly available, as are BAGPIPES[53], MultiNest[56], PyMultiNest[57], pPXF[60] and SpectRes[80]. BEAGLE[51] is available via a Docker image upon request from http://www.iap.fr/beagle/.

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

**Acknowledgements** This work is based on observations made with the NASA/ESA/CSA James Webb Space Telescope. The data were obtained from the Mikulski Archive for Space Telescopes at the Space Telescope Science Institute, which is operated by the Association of Universities for Research in Astronomy, Inc., under NASA contract NAS 5-03127 for JWST. These observations are associated with programmes 1180, 1210, 1895 and 1963. We acknowledge the FRESCO team led by PI P. Oesch for developing their observing programme with a zero-exclusive-access period. J.W. gratefully acknowledges support from the Fondation MERAC. J.W., R.M., M.C., F.D.E., T.J.L., W.M.B., L.S. and J.S. acknowledge support from the Science and Technology Facilities Council (STFC), the European Research Council (ERC) through Advanced Grant 695671, QUENCH, and a Frontier Research grant RISEandFALL from UK Research and Innovation. R.S. acknowledges support from an STFC Ernest Rutherford Fellowship (ST/S004831/1). R.M. also acknowledges funding from a research professorship from the Royal Society. S. Carniani acknowledges support from the European Union's Horizon Europe ERC Starting Grant No. 101040227 – WINGS. E.C.-L. acknowledges support from an STFC Webb Fellowship (ST/W001438/1). S. Arribas, B.R.d.P. and M.P. acknowledge support from the research project PID2021-127718NB-I00 of the Spanish Ministry of Science and Innovation/State Agency of Research (MICIN/AEI). A.J.B., A.J.C., J.C. and A.S. acknowledge funding from a FirstGalaxies Advanced Grant from the ERC under the European Union's Horizon 2020 research and innovation programme (grant agreement no. 789056). S. Alberts acknowledges support from the JWST Mid-Infrared Instrument Science Team Lead (grant no. 80NSSC18K0555) from NASA Goddard Space Flight Center to the University of Arizona. E.E., D.J.E., B.D.J., M.R., B.E.R., F.S. and C.N.A.W. acknowledge the JWST/NIRCam contract to the University of Arizona (ANAS5-02015). D.J.E. is supported as a Simons Investigator. M.P. acknowledges support from the Programa Atracción de Talento de la Comunidad de Madrid (grant no. 2018-T2/TIC-11715). C.C.W. is supported by NOIRLab, which is managed by the Association of Universities for Research in Astronomy under a cooperative agreement with the National Science Foundation. This research is supported in part by the Australian Research Council Centre of Excellence for All Sky Astrophysics in 3 Dimensions (ASTRO 3D), through project no. CE170100013. This study made use of the Prospero high-performance computing facility at Liverpool John Moores University. We acknowledge use of the lux supercomputer at the University of California, Santa Cruz, funded by the Major Research Instrumentation Program of the National Science Foundation (grant no. AST 1828315).

**Author contributions** J.W., I.S., R.S. and R.M. led the writing of this paper. M.R., E.E., F.S., K.N.H. and C.C.W. contributed to the design, construction and commissioning of NIRCam. A.J.B., C.N.A.W., C.W., D.J.E., M.R., P.F., R.M., S. Alberts and S. Arribas contributed to the design of the JADES survey. B.E.R., S.T., B.D.J., C.N.A.W., D.J.E., I.S., R.E., S.Alberts and Z.J. contributed to the JADES imaging data reduction. B.E.R. contributed to the JADES imaging data visualization. B.D.J., S.T., R.E., E.N. and W.M.B. contributed to the modelling of galaxy photometry. K.N.H., J.L. and R.E. contributed to the photometric redshift determination and target selection. B.E.R., C.N.A.W., C.C.W., K.N.H. and M.R. contributed to the JADES preflight imaging data challenges. S. Carniani, M.C., J.W., P.F., G.G., S. Arribas, M.P. and B.R.d.P. contributed to the NIRSpec data reduction and to the development of the NIRSpec pipeline. S. Arribas contributed to the design and optimization of the MSA configurations. A.J.C., A.J.B., C.N.A.W., E.C.-L. and K.B. contributed to the selection, prioritization and visual inspection of the targets. S. Charlot, J.C., E.C.-L., R.M., J.W., R.S., F.D.E., T.J.L., M.C., A.d.G., A.S. and L.S. contributed to analysis of the spectroscopic data, including redshift determination and spectral modelling. P.F., T.R., G.G., N.K. and B.R.d.P. contributed to the design, construction and commissioning of NIRSpec. F.D.E., T.J.L., M.C., B.R.d.P., R.M., S. Arribas and J.S. contributed to the development of the tools for the spectroscopic data analysis, visualization and fitting. C.W. contributed to the design of the spectroscopic observations and MSA configurations. B.E.R., C.W., D.J.E., M.R. and R.M. serve on the JADES Steering Committee.

**Competing interests** The authors declare no competing interests.

**Additional information**
**Correspondence and requests for materials** should be addressed to Joris Witstok, Irene Shivaei or Renske Smit.

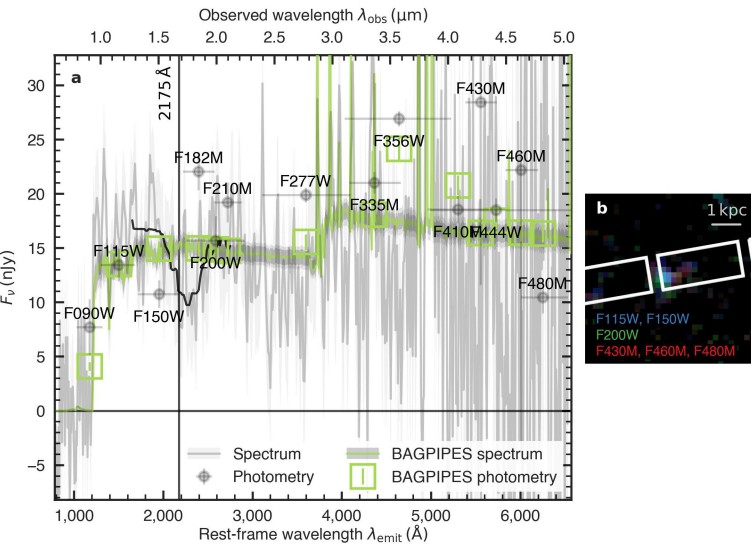

**Extended Data Fig. 1 | SED modelling and false-colour image of JADES-GS-z6-0. a**, Spectrum observed with NIRSpec (grey solid line and light-grey shading as $1\sigma$ uncertainty), overlaid with NIRCam photometry (grey points; error bars show the filter widths along the wavelength direction and $1\sigma$ uncertainty along the y-axis) of JADES-GS-z6-0. A calibration correction is applied to the spectrum to match the photometry (see the Stellar population synthesis modelling section). The running median around 2175 Å is shown as a solid black line. The best-fit bagpipes SED model (Stellar population synthesis modelling section) is shown as a light green solid line (predicted spectrum; darker and lighter shading represents the $1\sigma$ and $2\sigma$ uncertainty, respectively) and points (predicted photometry; error bars show $1\sigma$ uncertainty). **b**, Colour-composite $1'' \times 1''$ image constructed from inverse-variance weighted combinations of NIRCam filters, with the F115W and F150W filters as the blue channel, the F200W filter as green, and three medium-band filters not contaminated by strong emission lines (F430M, F460M, and F480M) as red. The position of the NIRSpec MSA shutters in the central nodding position (shown for all three separate observing visits, which however are nearly identical) are overlaid in white. A scale of 1 kpc is indicated in the bottom right.

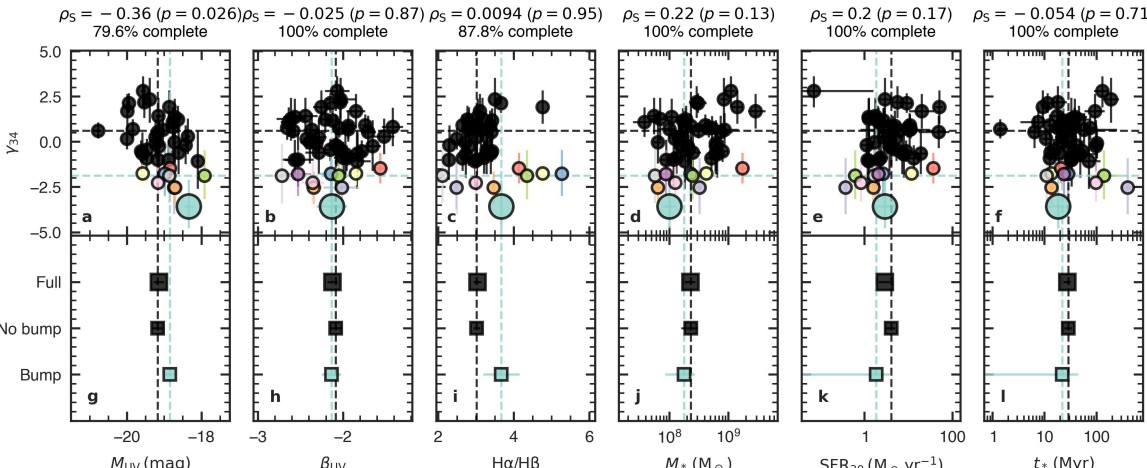

**Extended Data Fig. 2 | Sample characteristics.** Top row (panels **a** to **f**): spectral slope change around $\lambda_{emit} = 2175$ Å ($\gamma_{34}$) as a function of the physical properties of the galaxy sample. JADES-GS-z6-0 is highlighted as an enlarged point. Error bars represent $1\sigma$ uncertainty. The median values of galaxies belonging to the sample with (coloured points) and without (black points) a UV bump signature (see the Sample selection section) are indicated respectively with coloured and black dashed lines. The Spearman's rank correlation coefficient, $\rho_S$, is reported at the top of each panel along with its $p$-value and the completeness (i.e. for which percentage of the sample the metric was measured). Bottom row (panels **g** to **l**): median of the full sample (large black square), the sample with (coloured square) and without (small black square) bump signatures. The error bars show uncertainties on the median obtained with bootstrapping. Quantities shown are the absolute UV magnitude ($M_{UV}$; panels **a** and **g**), UV spectral slope ($\beta_{UV}$; **b** and **h**), Balmer decrement Hα/Hβ (**c** and **i**), stellar mass ($M_*$; **d** and **j**), star formation rate averaged over the last 30 Myr (SFR$_{30}$; **e** and **k**), and mass-weighted stellar age ($t_*$; **f** and **l**).

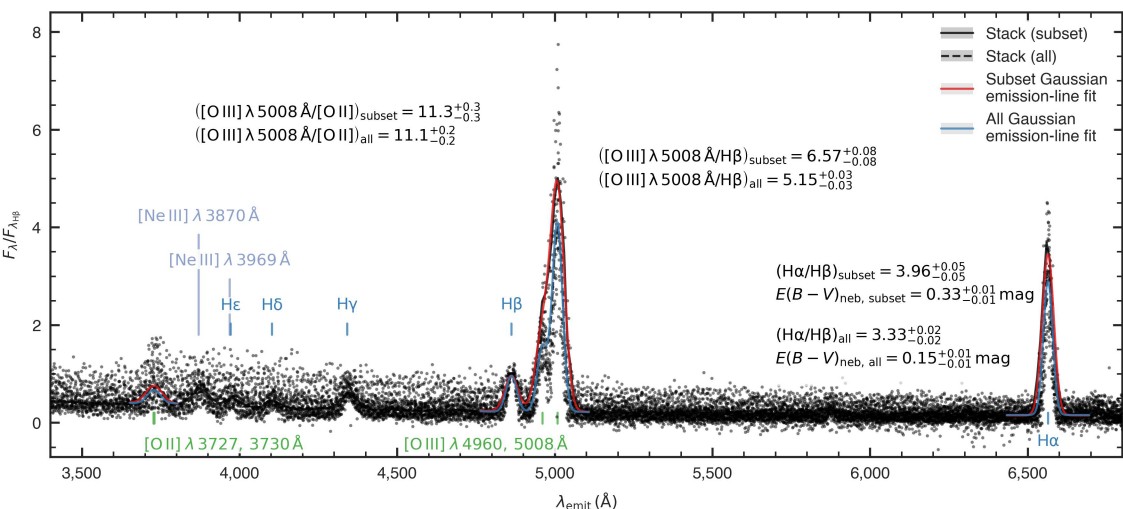

**Extended Data Fig. 3 | Normalised and stacked rest-frame optical spectra of z > 4 JADES galaxies observed by *JWST*/NIRSpec.** Similar to Fig. 2, spectra of all galaxies (small black dots) are shifted to the rest frame but normalised to the flux density at the rest-frame wavelength of Hβ, $\lambda_{H\beta}$ = 4862.7 Å. The dashed black line (shading represents 1σ uncertainty) shows a stacked spectrum obtained by combining all 49 objects in wavelength bins of $\Delta\lambda_{emit}$ = 10 Å. The main rest-frame optical emission lines are labelled. The stacked spectrum of ten galaxies selected to have a bump signature (solid black line, shading as 1σ uncertainty) exhibits stronger [O III] and Hα emission relative to the full sample, while having a consistent [O III]/[O II] line ratio. This is quantified by the integrated flux ratios of the fitted Gaussian line profiles (respectively shown as red and blue solid lines with shading as 1σ uncertainty), indicating differences in oxygen abundance as well as dust obscuration.

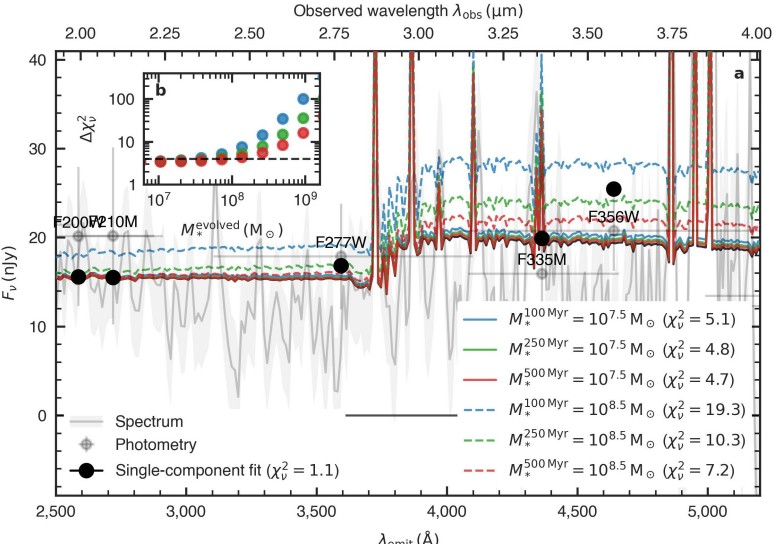

**Extended Data Fig. 4 | Exploration of signatures from substantially evolved stellar populations. a**, The age-sensitive spectral region around the 4000 Å (Balmer) break of JADES-GS-z6-0. Similar to Extended Data Fig. 1, the (photometry-corrected) NIRSpec spectrum is shown as the grey solid line (light-grey shading as $1\sigma$ uncertainty), NIRCam photometry as grey points (error bars show the filter widths along the wavelength direction and $1\sigma$ uncertainty along the y-axis). The BAGPIPES SED model with best-fit parameters (see Stellar population synthesis modelling) is shown as a black solid line (spectrum) and points (photometry). Two-component models are shown by the blue (age of 100 Myr), green (250 Myr), and red (500 Myr) solid ($10^{7.5}$ M$_\odot$) and dashed ($10^{8.5}$ M$_\odot$) lines. **b**, The difference in reduced chi-squared values, $\Delta\chi_\nu^2$, is shown as a function of the stellar mass of the evolved population, $M_{evolved}$. A dashed horizontal line indicates the value above which the new model is in $2\sigma$ tension with respect to the single-component model ($\Delta\chi_\nu^2 = 4$).

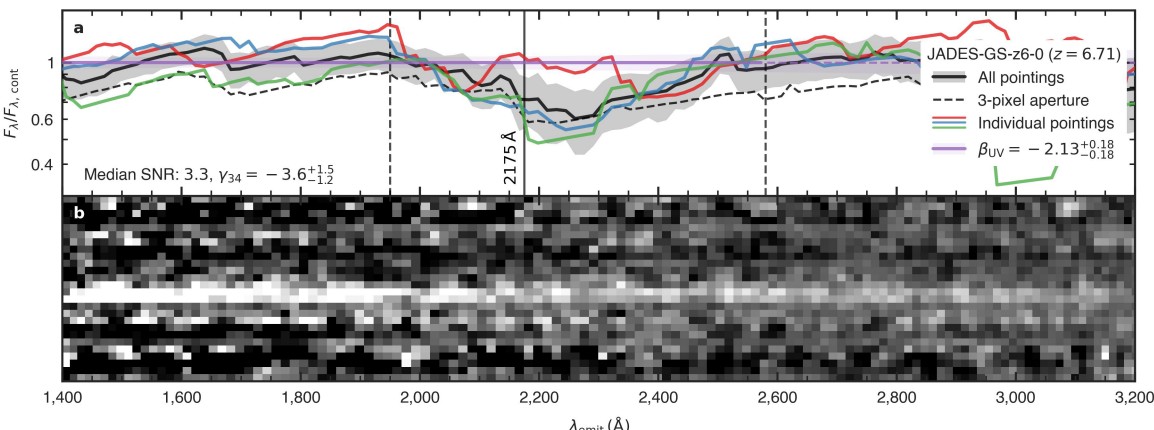

**Extended Data Fig. 5 | Rest-frame UV continuum of JADES-GS-z6-0. a**, The one-dimensional spectrum of JADES-GS-z6-0 (smoothed with a 15-pixel median filter as in Fig. 1) is normalised to the predicted continuum level in the absence of a UV bump, modelled as a power law with index $\beta_{UV}$ (purple line; shading as $1\sigma$ uncertainty). The median signal-to-noise ratio (SNR) and $\gamma_{34}$ are reported in the bottom-left corner. Coloured lines show data from individual observing visits, while the solid black line and grey shading represent the combined spectra and their $1\sigma$ uncertainty, respectively. A dashed black line indicates the spectrum from a 3-pixel aperture extraction. **b**, Two-dimensional spectrum of JADES-GS-z6-0 (not scaled to the predicted continuum level).

**Extended Data Table 1 | Properties of the galaxies in the bump sample**

| Source name | $z_{\text{spec}}$ | $m_{\text{F444W}}$ (mag) | $M_{\text{UV}}$ (mag) | $\beta_{\text{UV}}$ | $H\alpha/H\beta$ |
|---|---|---|---|---|---|
| JADES-GS+53.15138-27.81917[†] | $6.7065^{+0.0004}_{-0.0003}$ | $28.23 \pm 0.06$ | $-18.34 \pm 0.12$ | $-2.13^{+0.18}_{-0.18}$ | $3.67 \pm 0.24$ |
| JADES-GS+53.16871-27.81516[‡] | $4.0223^{+0.0002}_{-0.0002}$ | $26.17 \pm 0.01$ | $-19.58 \pm 0.06$ | $-1.85^{+0.06}_{-0.06}$ | $4.76 \pm 0.07$ |
| JADES-GS+53.13059-27.80771 | $5.6166^{+0.0009}_{-0.0007}$ | $27.88 \pm 0.04$ | $-18.74 \pm 0.07$ | $-2.02^{+0.20}_{-0.21}$ | $2.48 \pm 0.41$ |
| JADES-GS+53.13284-27.80186[‡] | $4.6617^{+0.0001}_{-0.0001}$ | $26.71 \pm 0.01$ | $-18.85 \pm 0.02$ | $-1.56^{+0.09}_{-0.08}$ | $4.13 \pm 0.05$ |
| JADES-GS+53.17022-27.77739[‡] | $4.7094^{+0.0005}_{-0.0004}$ | $27.21 \pm 0.03$ | $-18.99 \pm 0.04$ | $-2.14^{+0.14}_{-0.14}$ | $5.27 \pm 0.76$ |
| JADES-GS+53.12689-27.77689 | $4.8878^{+0.0002}_{-0.0002}$ | $28.38 \pm 0.04$ | $-18.71 \pm 0.04$ | $-2.34^{+0.10}_{-0.09}$ | $3.46 \pm 0.33$ |
| JADES-GS+53.16743-27.77548[‡] | $4.1426^{+0.0002}_{-0.0002}$ | $27.55 \pm 0.02$ | $-17.91 \pm 0.08$ | $-2.05^{+0.16}_{-0.16}$ | $4.35 \pm 0.51$ |
| JADES-GS+53.16660-27.77240 | $6.3296^{+0.0002}_{-0.0002}$ | $27.98 \pm 0.05$ | $-19.17 \pm 0.06$ | $-2.35^{+0.13}_{-0.13}$ | $2.99 \pm 0.11$ |
| JADES-GS+53.13423-27.76891 | $7.0493^{+0.0003}_{-0.0003}$ | $28.12 \pm 0.02$ | $-18.86 \pm 0.03$ | $-2.72^{+0.17}_{-0.18}$ | $2.11 \pm 0.14**$ |
| JADES-GS+53.11833-27.76901 | $7.2043^{+0.0002}_{-0.0002}$ | $27.77 \pm 0.08$ | $\ldots$ * | $-2.53^{+0.15}_{-0.15}$ | $\ldots$ ** |

Error bars on measurements represent 1σ uncertainty. Rows: (1) Source name (including the J2000 RA and Dec in deg), (2) Spectroscopic redshift ($z_{\text{spec}}$), (3) Apparent AB magnitude in the NIRCam F444W filter ($m_{\text{F444W}}$), (4) Absolute AB magnitude in the UV ($M_{\text{UV}}$), (5) UV spectral slope ($\beta_{\text{UV}}$), (6) Balmer decrement (Hα/Hβ).

[†]JADES-GS-z6-0.

[‡]Contained in the blind selection discussed in the Robustness of the UV bump detections section.

*This source falls outside the main NIRCam footprint.

**At this redshift, Hα (partially) shifts outside of the NIRSpec coverage.

**Extended Data Table 2 | Properties of the samples of galaxies with and/or without indication of a UV bump**

|  | Bump sample | Non-bump sample | Full sample |
|---|---|---|---|
| *Individual properties* | | | |
| $z_{\mathsf{spec}}$ | $5.25 \pm 0.68$ | $5.77 \pm 0.27$ | $5.62 \pm 0.29$ |
| $M_{\mathsf{UV}}$ (mag) | $-18.85 \pm 0.14$ | $-19.17 \pm 0.13$ | $-19.14 \pm 0.13$ |
| $\beta_{\mathsf{UV}}$ | $-2.14 \pm 0.12$ | $-2.08 \pm 0.07$ | $-2.12 \pm 0.06$ |
| $\gamma_{34}$ | $-1.86 \pm 0.21$ | $0.62 \pm 0.26$ | $-0.03 \pm 0.34$ |
| $Z_{\mathsf{neb}}$ ($\mathrm{Z_\odot}$) | $0.17 \pm 0.05$ | $0.10 \pm 0.02$ | $0.11 \pm 0.02$ |
| $E(B-V)_{\mathsf{neb}}$ (mag) | $0.25 \pm 0.13$ | $0.06 \pm 0.03$ | $0.06 \pm 0.03$ |
| $M_*$ ($10^8\ \mathrm{M_\odot}$) | $1.8 \pm 0.8$ | $2.3 \pm 0.7$ | $2.3 \pm 0.4$ |
| $Z_*$ ($\mathrm{Z_\odot}$) | $0.06 \pm 0.03$ | $0.17 \pm 0.05$ | $0.13 \pm 0.04$ |
| $\mathsf{SFR}_{30}$ ($\mathrm{M_\odot\ yr^{-1}}$) | $1.9 \pm 1.6$ | $4.1 \pm 1.0$ | $3.0 \pm 0.8$ |
| $t_*$ (Myr) | $22 \pm 24$ | $29 \pm 5$ | $27 \pm 4$ |
| *Stack properties* | | | |
| $N_{\mathsf{sample}}$ | $10$ | $39$ | $49$ |
| $\beta$ | $-1.95^{+0.02}_{-0.02}$ | $\ldots$ | $-2.18^{+0.01}_{-0.01}$ |
| Hα/Hβ | $3.96^{+0.05}_{-0.05}$ | $\ldots$ | $3.33^{+0.02}_{-0.02}$ |
| $E(B-V)_{\mathsf{neb}}$ (mag) | $0.33^{+0.01}_{-0.01}$ | $\ldots$ | $0.15^{+0.01}_{-0.01}$ |
| [O III] λ 5008 Å/Hβ | $6.57^{+0.08}_{-0.08}$ | $\ldots$ | $5.15^{+0.03}_{-0.03}$ |
| [O III] λ 5008 Å/[O II] | $11.3^{+0.3}_{-0.3}$ | $\ldots$ | $11.1^{+0.2}_{-0.2}$ |

Properties of the samples are presented as median values; error bars represent a 1σ uncertainty (obtained with bootstrapping for the median properties). Rows of individual properties: (1) Spectroscopic redshift ($z_{\mathsf{spec}}$), (2) Absolute AB magnitude in the UV ($M_{\mathsf{UV}}$), (3) UV spectral slope ($\beta_{\mathsf{UV}}$), (4) Spectral slope change around $\lambda_{\mathsf{emit}}$=2175 Å ($\gamma_{34}$), (5) Gas-phase metallicity ($Z_{\mathsf{neb}}$; from rest-frame optical emission lines) in units of Solar metallicity, (6) Nebular extinction ($E(B-V)_{\mathsf{neb}}$; from the Balmer decrement) in magnitudes, (7) Stellar mass ($M_*$) in $10^8$ Solar masses, (8) Stellar metallicity ($Z_*$; from SED modelling) in units of Solar metallicity, (9) Star formation rate in Solar masses per year averaged on a timescale of 30 Myr ($\mathsf{SFR}_{30}$), (10) Mass-weighted stellar age ($t_*$) in Myr. Rows of properties measured in stacked spectra: (1) Number of galaxies contained within the sample ($N_{\mathsf{sample}}$), (2) Power-law slope ($\beta$), (3) Balmer decrement Hα/Hβ, (4) Nebular extinction ($E(B-V)_{\mathsf{neb}}$) in magnitudes, (5) [O III] λ 5008 Å/Hβ line ratio, (6) [O III] λ 5008 Å/[O II] line ratio.