## [Peer Review File · Nature]

Manuscript Title: Carbonaceous dust grains seen in the first billion years of cosmic time

Reviewer Comments & Author Rebuttals

Reviewer Reports on the Initial Version:

Referees' comments:

Referee #1 (Remarks to the Author):

A. The authors report the first detection of the 2175Å UV bump, currently understood to arise due to carbonaceous interstellar dust grains, in galaxies with cosmological redshifts that exceed a value of 3. In particular, the detection of this absorption feature in the spectrum of a galaxy at redshift 6.71 presents a challenge for the idea that AGB stars are the primary source of such grains because of the implied amount of dust required at this early epoch of galactic evolution (about 600 million years following the Big Bang).

B. I think that this work represents a significant contribution to the empirical data on dust and chemical evolution in the early universe, effectively halving the temporal distance between the reported observations and the beginning of the universe and approaching the limit at which galaxy formation is thought to occur. This data should help shape galactic evolutionary theory.

C. In my judgement, the data is analyzed in a robust fashion, making use of the diverse expertise of the contributing authors. The analysis is detailed but concise, ably addressing the complexity of the data acquisition and processing, and the implications of the results. The background material is well-summarised and potential challenges to the implications raised, in terms of the age of the stellar population in the sample galaxies, are directly addressed.

D. The statistical tests used in the data's analysis are appropriate; the errors are reliably presented in the body of the article and in tables and figures – giving readers the opportunity to judge the robustness of the findings for themselves, with a high degree of assurance.

E. I find the conclusions to be robust, convincing, and compelling – I think that this article will be cited often for the extent to which it provides meaningful empirical data on dust and chemical evolution in the early universe.

F. The one minor quibble I note is that the authors mention that the central wavelength of the UV bump apparent in the spectrum of JADES-GS+53.15139-27.81917 is at the high end of the range reported in Galactic features by reference [26]. More precisely, I note that the reported wavelength is nearly 2σ larger than the highest value published in [26]. This raises a question in my mind about whether there is a change in the distribution of bump properties in the early universe, particularly if AGB stars are no longer the primary mechanism of carbonaceous grain production. I note the article's references [10] and [35]; [10] also reports bump features centred at wavelengths near or longer than the Galactic values of [26] for galaxies near a redshift of 2. [35] concludes that the UV bump width is highly variable, in contrast with the mildly variable central wavelength. Notably, the bump positions reported here among sample galaxies is much closer to the mean of Galactic values, but the implication of some early-universe bump wavelengths well outside of the distribution peak of Galactic positions seems suggestive to me. I wonder if the authors would comment, or if that should be left for a time when there is more data.

G. The number and scope of references cited in this work are appropriate to the topic.

H. I am impressed by the quality of writing in this article. I think it is well-written – the abstract,

introductory material, and conclusions are all relevant and well-composed to describe the data being presented.

Referee #2 (Remarks to the Author):

This manuscript reports an exciting discovery of the mysterious 2175 Angstrom extinction bump in a high-redshift galaxy ($z=6.7$), based on the JWST/NIRSpec spectroscopic observations. The authors did an excellent job in carefully performing data reduction and analysis and the detection was robust and convincing, The analysis was also extensive, robust and convincing. This discovery is scientifically significant and will have important broad implication for a wide variety of astrophysical and astrochemical topics. The 2175 Angstrom extinction bump is the strongest spectral feature in the interstellar extinction curve. Its carrier remains unknown ever since its first discovery in 1965. The detection of this mysterious extinction bump at a distant galaxy when the universe was considerably less than 1 Gyr old will provide important insight into the origin and nature of the bump carrier, and more generally, on the dust production in the early universe. This will also offer insight into the formation and evolution of stars in the early universe and the cosmic star formation history. I believe that this manuscript should be published rapidly in Nature and it will stimulate many subsequent (theoretical and observational) works.

The manuscript is well written, concise (and comprehensive). I only have one question: what is the "excess" emission just at wavelength shortward of 2000 Angstrom which is in excess of the power-law UV continuum (see Figure 1) ? Or, there is actually no excess emission at <2000 Angstrom, the actual UV continuum is higher? if this is true, the actual 2175 Angstrom bump could be more prominent.

A relatively minor issue: some references are not the most relevant ones, e.g., refs. 3-7: perhaps just keep ref. 5; refs.8-13: perhaps just keep a couple of references. But these can be taken care after it is revised.

When it was mentioned "... is attributed to carbonaceous grains". I should point out that it was Li & Draine (2001) who first explicitly considered PAHs as the carrier of the 2175 Angstrom bump.

When it was mentioned "... Beyond the local Universe, the feature has previously only been observed in the spectra of massive, metal-enriched galaxies ...". Liang & Li (2009, ApJ, 690, L56) inferred the presence of the 2175 Angstrom extinction bump in the host galaxy of a gamma-ray burst (GRB) at $z=6.29$, based on photometric data. The extinction bump was also reported in nearer GRB hosts of $z<2$ (Liang & Li (2010, ApJ, 710, 648). Spectroscopically, Elíasdóttir et al. (2009, ApJ, 697, 1725) reported the detection of the 2175 Angstrom bump in a $z=2.45$ GRB.

Near the end of the main text, when discussing supernova origin of carbon dust at high z , there is actually another piece of evidence supporting the authors' idea: presolar graphite grains have been identified in primitive meteorites and isotope analysis reveals they were from Type II supernovae. In this context, graphite grains could be abundant in the early universe when Type II supernovae are the major dust producers and small graphitic grains cause the 2175 Angstrom extinction bump.

All these are minor and easy to take care. I believe this work represents an important breakthrough brought by JWST and will impact a wide range of subfields.

Referee #3 (Remarks to the Author):

I have reviewed the manuscript "Carbonaceous dust grains within galaxies seen in the first billion

years of cosmic time."

The authors report the possible discovery of a broad UV absorption feature at 2175 Angstroms in the spectra of 11 star-forming galaxies at $z > \sim 4$, with this feature appearing to be particularly significant in one star-forming galaxy at $z \sim 6.8$.

If these features are bona-fide and represent what the authors suggest, this would indeed be a very exciting discovery, and provide important insight into the composition of dust just ~ 1 billion years after the Big Bang -- and in my opinion be more than worthy of having the result appear in Nature.

Nevertheless, I am a bit worried about whether the features in the JWST spectra indeed represent what the authors suggest.

The sources that make up the "bump" sample do not appear to be particularly luminous or massive. Their colors are also not very red. At lower redshift, most UV faint or lower mass sources do not show substantial amounts of dust extinction.

What evidence do the authors have that the light in their "bump" sample is subject to significant dust extinction except for the presence of an apparent UV bump feature? No mention is made of ALMA coverage of their selected sample or whether Balmer decrement measurements are available. Do any sources in their "bump" sample show any significant dust continuum emission? As most of their sources seem to be located over GOODS South, I would expect archival ALMA observations to be available over most of their sources.

To the same point, is there any evidence for significant dust absorption in the sources based on their Balmer decrements? Assuming as a baseline "bump" sources found at lower redshifts, might the authors expect to see significant dust continuum emission / absorption in their "bump" sample?

How similar are sources in the "bump" sample to sources showing UV absorption features in lower redshift samples? Are the average extinction properties (A_V) and stellar masses similar to sources at lower redshifts? If so, it would greatly strengthen the paper to indicate these similarities (or differences).

I am a bit worried that the "bump" sample the authors report just happens to have peculiar features in their spectrum due to some other phenomenon and/or due to peculiarities with the spectral reductions of a few sources, and it might bear no relation to the phenomenon at lower redshift.

To make the evidence for finding this feature more compelling, could the authors repeat a search for such a "bump" feature but shifting the search 300 Angstroms lower or higher in wavelength? i.e., at 1875 A or 2475 A? How many sources do the authors find searching for a similar feature but at slightly lower or higher wavelength?

Additionally, if the authors split the spectral data they use to identify the "bump" sources into two halves and select "bump" sources using only the first half of the data, does a stack of the second half of the data show the same "bump" feature?

For transparency and reproducibility, it would be useful if the authors could provide the coordinates, redshifts, and other properties of the 11 galaxies that make up their "bump" sample in the supplementary information. How many sources are in the comparison sample?

Author Rebuttals to Initial Comments:

Referee #1 (Remarks to the Author):

A. The authors report the first detection of the 2175Å UV bump, currently understood to arise due to carbonaceous interstellar dust grains, in galaxies with cosmological redshifts that exceed a value of 3. In particular, the detection of this absorption feature in the spectrum of a galaxy at redshift 6.71 presents a challenge for the idea that AGB stars are the primary source of such grains because of the implied amount of dust required at this early epoch of galactic evolution (about 600 million years following the Big Bang).

B. I think that this work represents a significant contribution to the empirical data on dust and chemical evolution in the early universe, effectively halving the temporal distance between the reported observations and the beginning of the universe and approaching the limit at which galaxy formation is thought to occur. This data should help shape galactic evolutionary theory.

C. In my judgement, the data is analyzed in a robust fashion, making use of the diverse expertise of the contributing authors. The analysis is detailed but concise, ably addressing the complexity of the data acquisition and processing, and the implications of the results. The background material is well-summarised and potential challenges to the implications raised, in terms of the age of the stellar population in the sample galaxies, are directly addressed.

D. The statistical tests used in the data's analysis are appropriate; the errors are reliably presented in the body of the article and in tables and figures – giving readers the opportunity to judge the robustness of the findings for themselves, with a high degree of assurance.

E. I find the conclusions to be robust, convincing, and compelling – I think that this article will be cited often for the extent to which it provides meaningful empirical data on dust and chemical evolution in the early universe.

F. The one minor quibble I note is that the authors mention that the central wavelength of the UV bump apparent in the spectrum of JADES-GS+53.15139-27.81917 is at the high end of the range reported in Galactic features by reference [26]. More precisely, I note that the reported wavelength is nearly 2σ larger than the highest value published in [26]. This raises a question in my mind about whether there is a change in the distribution of bump properties in the early universe, particularly if AGB stars are no longer the primary mechanism of carbonaceous grain production. I note the article's references [10] and [35]; [10] also reports bump features centred at wavelengths near or longer than the Galactic values of [26] for galaxies near a redshift of 2. [35] concludes that the UV bump width is highly variable, in contrast with the mildly variable central wavelength. Notably, the bump positions reported here among sample galaxies is much closer to the mean of Galactic values, but the implication

of some early-universe bump wavelengths well outside of the distribution peak of Galactic positions seems suggestive to me. I wonder if the authors would comment, or if that should be left for a time when there is more data.

- We agree with the referee that the seemingly longer central wavelength is worth pointing out. We include a short, speculative comment stating that this may point towards a different nature of the grains responsible for the absorption both in the main text and at the end of the summary paragraph. This question, however, will certainly be investigated further in future work exploiting larger samples from JADES and other programs.

G. The number and scope of references cited in this work are appropriate to the topic.

H. I am impressed by the quality of writing in this article. I think it is well-written – the abstract, introductory material, and conclusions are all relevant and well-composed to describe the data being presented.

Referee #2 (Remarks to the Author):

This manuscript reports an exciting discovery of the mysterious 2175 Angstrom extinction bump in a high-redshift galaxy ($z=6.7$), based on the JWST/NIRSpec spectroscopic observations. The authors did an excellent job in carefully performing data reduction and analysis and the detection was robust and convincing. The analysis was also extensive, robust and convincing. This discovery is scientifically significant and will have important broad implications for a wide variety of astrophysical and astrochemical topics. The 2175 Angstrom extinction bump is the strongest spectral feature in the interstellar extinction curve. Its carrier remains unknown ever since its first discovery in 1965. The detection of this mysterious extinction bump at a distant galaxy when the universe was considerably less than 1 Gyr old will provide important insight into the origin and nature of the bump carrier, and more generally, on the dust production in the early universe. This will also offer insight into the formation and evolution of stars in the early universe and the cosmic star formation history. I believe that this manuscript should be published rapidly in Nature and it will stimulate many subsequent (theoretical and observational) works.

The manuscript is well written, concise (and comprehensive). I only have one question: what is the "excess" emission just at wavelength shortward of 2000 Angstrom which is in excess of the power-law UV continuum continuum (see Figure 1) ? Or, there is actually no excess emission at <2000 Angstrom, the actual UV continuum is higher? if this is true, the actual 2175 Angstrom bump could be more prominent.

- Firstly, there is significant emission from the CIII doublet (both in the individual spectrum of JADES-GS-z6-0 and the stacked spectra) at a rest-frame wavelength of ~ 1910 Å which, as discussed in the Methods, is explicitly excluded when fitting the power-law UV continuum with the Calzetti windows (now indicated by the hatched region). Close to the CIII doublet, there was some additional weak excess emission. This however has become quite minor in the new reduction and is mostly removed when considering the running median, suggesting this excess was due to potential minor data artefacts (e.g. cosmic rays); we note the inferred bump amplitude remains mostly unchanged in the new data reduction.

A relatively minor issue: some references are not the most relevant ones, e.g., refs. 3-7: perhaps just keep ref. 5; refs. 8-13: perhaps just keep a couple of references. But these can be taken care of after it is revised.

- We have adopted these suggestions.

When it was mentioned "... is attributed to carbonaceous grains". I should point out that it was Li & Draine (2001) who first explicitly considered PAHs as the carrier of the 2175 Angstrom bump.

- We have included a reference to the work of Li & Draine (2001) when discussing PAHs as the origin of the 2175 Å absorption feature.

When it was mentioned "... Beyond the local Universe, the feature has previously only been observed in the spectra of massive, metal-enriched galaxies ...". Liang & Li (2009, ApJ, 690, L56) inferred the presence of the 2175 Angstrom extinction bump in the host galaxy of a gamma-ray burst (GRB) at $z=6.29$, based on photometric data. The extinction bump was also reported in nearer GRB hosts of $z < 2$ (Liang & Li (2010, ApJ, 710, 648). Spectroscopically, Elíasdóttir et al. (2009, ApJ, 697, 1725) reported the detection of the 2175 Angstrom bump in a $z=2.45$ GRB.

- We have adjusted the wording to reflect the fact that GRB absorption reflects the dust extinction along a single sightline (as discussed in the Bump amplitude comparison section), rather than the attenuation of the integrated spectrum of a galaxy. Moreover, we have referenced the work of Liang & Li (2009, 2010) and Elíasdóttir et al. (2009).

Near the end of the main text, when discussing supernova origin of carbon dust at high z , there is actually another piece of evidence supporting the authors' idea: presolar graphite grains have been identified in primitive meteorites and isotope analysis reveals they were from Type II supernovae. In this context, graphite grains could be abundant in the early universe when Type II supernovae are the major dust producers and small graphitic grains cause the 2175 Angstrom extinction bump.

- We agree that this is a very relevant line of evidence that we have added to the discussion.

All these are minor and easy to take care. I believe this work represents an important breakthrough brought by JWST and will impact a wide range of subfields.

Referee #3 (Remarks to the Author):

I have reviewed the manuscript "Carbonaceous dust grains within galaxies seen in the first billion years of cosmic time."

The authors report the possible discovery of a broad UV absorption feature at 2175 Angstroms in the spectra of 11 star-forming galaxies at $z > \sim 4$, with this feature appearing to be particularly significant in one star-forming galaxy at $z \sim 6.8$.

If these features are bona-fide and represent what the authors suggest, this would indeed be a very exciting discovery, and provide important insight into the composition of dust just ~ 1 billion years after the Big Bang -- and in my opinion be more than worthy of having the result appear in Nature.

Nevertheless, I am a bit worried about whether the features in the JWST spectra indeed represent what the authors suggest.

The sources that make up the "bump" sample do not appear to be particularly luminous or massive. Their colors are also not very red. At lower redshift, most UV faint or lower mass sources do not show substantial amounts of dust extinction.

What evidence do the authors have that the light in their "bump" sample is subject to significant dust extinction except for the presence of an apparent UV bump feature? No mention is made of ALMA coverage of their selected sample or whether Balmer decrement measurements are available. Do any sources in their "bump" sample show any significant dust continuum emission? As most of their sources seem to be located over GOODS South, I would expect archival ALMA observations to be available over most of their sources.

- Indeed, archival ALMA observations are available over GOODS-S. We have investigated the archive and discuss our findings in a new section in the methods ("Ancillary far-infrared observations"). In brief, we do not find any evidence for individually detected galaxies, nor dust-continuum detections in stacked images; we discuss that these galaxies are simply too faint to be detected in the available (rather

shallow) ALMA data, even if they had relatively high dust-to-stellar mass ratios, so that unfortunately the ALMA data are not constraining.

To the same point, is there any evidence for significant dust absorption in the sources based on their Balmer decrements? Assuming as a baseline "bump" sources found at lower redshifts, might the authors expect to see significant dust continuum emission / absorption in their "bump" sample?

- We agree with the referee that the sources in our bump sample should be expected as a minimum to show hints of increased dust obscuration. We have investigated this in several different ways as discussed extensively in the various additions to the methods (mainly in the "Robustness of the UV bump detections" section), including via the Balmer decrement measured both in individual as well as stacked spectra. We do indeed find evidence for an enhanced Balmer decrement (suggesting 2x higher E(B-V) in the bump sample compared to the full sample), mild reddening of the continuum slope (-1.95 versus -2.12 typical of the full sample), and a hint of higher gas-phase metallicity in the sources contained in the bump sample.

How similar are sources in the "bump" sample to sources showing UV absorption features in lower redshift samples? Are the average extinction properties (A_V) and stellar masses similar to sources at lower redshifts? If so, it would greatly strengthen the paper to indicate these similarities (or differences).

- The measured extinction properties (discussed above) are now put into context of known bump samples at lower redshift, both in the main text and the relevant methods section. We find the extinction to be similar to what was found by Noll et al. (2009), while stellar masses of our sample are significantly lower, as is discussed with Fig. 3.

I am a bit worried that the "bump" sample the authors report just happens to have peculiar features in their spectrum due to some other phenomenon and/or due to peculiarities with the spectral reductions of a few sources, and it might bear no relation to the phenomenon at lower redshift.

To make the evidence for finding this feature more compelling, could the authors repeat a search for such a "bump" feature but shifting the search 300 Angstroms lower or higher in wavelength? i.e., at 1875 Å or 2475 Å? How many sources do the authors find searching for a similar feature but at slightly lower or higher wavelength?

- We have carried out this test at a wavelength of 2475 Å to avoid contamination of UV emission lines. We find that although an absorption feature appears, it is weaker and narrower than the observed bump signature at 2175 Å and shows positive flux excess at the edges of the window, therefore producing a poor fit (having much lower amplitude) when fitted with a Drude profile with FWHM of 250 Å (see figure below). In short, this stack of noisy signals is produced by a combination of both positive (at the edges of our window) and negative (at the 2475 Å centre of the window) noise features, affecting one or a few pixels for each noise feature (i.e. noise patterns expected from detector noise or cosmic rays impacting one or a few pixels on the detector). Such a narrow noise feature is evidently different from that seen in our 2175 Å selected sample. We conclude that the likelihood of producing a broad feature of 250 Å at rest-frame 2175 Å (~8 native detector pixels for a source at $z = 5$) in a sample of 10 galaxies solely due to noise is extremely unlikely. Moreover, sources selected to have a continuum inflection around this wavelength do not show the same enhanced $H\alpha/H\beta$ or $[OIII]/H\beta$ ratios as seen in the y_{34} -selected bump sample. We discuss the outcome in the new "Robustness" section.

Additionally, if the authors split the spectral data they use to identify the "bump" sources into two halves and select "bump" sources using only the first half of the data, does a stack of the second half of the data show the same "bump" feature?

- While the bump signature in the stacked spectrum is pushing the signal-to-noise ratio we can achieve with the available data, we have verified that randomly splitting the bump sample into two halves yields similar results (shown in the first figure below). We further found that stringent criteria on the (highest) Balmer decrements and UV slopes in the parent sample select four sources all contained within the bump sample, whose significantly reddened (slope of -1.69) stacked spectrum reveals a tentative detection of the bump feature (second figure below). This "blind" sample is also highlighted in the new table with properties of the bump sample galaxies (see comment below). We again describe these findings in the "Robustness" section.

For transparency and reproducibility, it would be useful if the authors could provide the coordinates, redshifts, and other properties of the 11 galaxies that make up their "bump" sample in the supplementary information. How many sources are in the comparison sample?

- We agree that it clearly benefits the paper to specify the coordinates, redshifts, and other properties of these galaxies, which we have included in a new table in the Methods section. The full sample comprises 49 galaxies (in the previous version of the data reduction this was 50), meaning that there are 39 galaxies not belonging to the bump sample.

Reviewer Reports on the First Revision:

Referees' comments:

Referee #1 (Remarks to the Author):

The authors have addressed the points I raised in my review of their first draft. I recommend that this revision be published. Congratulations!

Referee #2 (Remarks to the Author):

The revised version satisfactorily addressed my concerns. I (and the potential readers as well, I believe) appreciate the authors' careful and comprehensive data reduction and statistical analysis. The detection of the 2175 Angstrom extinction bump at $z=6.7$ is robust and will be of big impacts in a range of scientific topics. I am happy to recommend the acceptance of this manuscript for rapid publication in Nature.

Regarding the relatively long central wavelength of the extinction bump (at 2263 Angstrom, instead of the canonical 2175 Angstrom), I believe, it may be related to the physical conditions. Blasberger et al. (2017, ApJ, 836, 173) reported that the interstellar extinction bump of some lines of sight toward some Herbig Ae/Be stars peaks at substantially longer wavelength (see their Table 1). The exact reason remains unknown, probably related to different mixtures of PAHs (or ultrasmall aromatic grains like graphitic nanoparticles). So, the fact that at $z=6.7$ the central wavelength of the 2175 Angstrom extinction bump peaks at a longer wavelength actually is itself interesting and will stimulate interest in laboratory astrophysics studies of cosmic dust analogues (like carbon onions, PAHs etc, see Li et al. 2008, MNRAS, 390, L39). The authors may add a sentence or so on this. But it is also fine if the authors decide not to discuss. I am pretty sure that shortly theorists and experimentalists will explore this issue.

There are several very minor issues:

Abstract:

``... often thought to be polycyclic aromatic hydrocarbons (PAHs) produced on times scales ..." --
> ``... often thought to be polycyclic aromatic hydrocarbons (PAHs) nano-sized graphitic grains produced on times scales ..."

Main text, 2nd paragraph, ``... and is attributed to carbonaceous dust grains, specifically PAHs or graphite^{1,16}." Reference #16 is not very relevant. I suggest to refer to #9 and #15, instead of #1 and #16. Actually, #16 could be deleted.

Main text, 3rd paragraph, ``... furthermore suggest JADES-GS-z6-0 has undergone ...",
furthermore  further

References:

#10 (Cox et al.) not very relevant, could be deleted.

#27 Draine, B.T. 1989, not ``Draine, Allamandola, Tielens (eds)". This was a review article by Draine alone, published in an IAU Symp. Proceedings edited by Allamandola & Tielens

There are some math symbols did not show up in the references,
e.g., #8, ..of $z > 7$ Lyman break galaxies
#13, ... at redshifts $z > 10$...

#18, ... at z 2...
#28, ... at 0.1 z < 3

...

I believe that these were caused by latex and could be easily fixed by adding \$..\$.

Referee #3 (Remarks to the Author):

I thank the authors for responding so completely to my previous feedback and concerns. I was particularly impressed by the following point the authors make in their updated methods section:

"Finally, we have verified that we are able to perform a blind selection from the parent sample of sources with the highest Balmer decrements and reddest UV slopes which results in a tentative detection of the UV bump. Specifically, requiring a Balmer decrement of $H\alpha/H\beta \gtrsim 4$ and UV slope of $\beta_{UV} \gtrsim -2.2$ yields a sample of four sources all contained within the bump sample (i.e. JADES-GS+53.16871-27.81516, JADES-GS+53.13284-27.80186, JADES-GS+53.17022-27.77739, and JADES-GS+53.16743-27.77548; see Table 1) but notably excludes JADES-GS-z6-0."

I only have a few minor questions and suggestions, which I list below:

1) The measured Halpha/Hbeta ratio for JADES-GS+53.13423-27.76891 at z=7.049 is 2.11 +/- 0.14, 5 sigma below the intrinsic ratio 2.86. Could the Balmer decrement measurement be impacted by the fact that the Halpha line lies at ~5.3 microns close to where the sensitivity of NIRSPEC cuts off? Is this worth remarking on?

2) While I appreciated the authors providing a brief discussion of the available far-IR ALMA constraints for sources, I think it would improve the paper slightly if the current text in the Methods section could be made a bit more quantitative. Are ALMA observations available for every source in their "bump" sample -- or are there some sources with no coverage? Is ALMA coverage available from only one program (ASAGAO) or also from other programs, e.g., ALMA GOODS (PI: Elbaz) or ASPECS (PI: Walter)? Also could the authors list the data set-dependent continuum sensitivities available to probe the dust-continuum emission from sources in their "bump" sample?

Outside of the two minor issues listed above, I find the presentation to be reasonably convincing overall and so am happy to recommend to Nature to publish the result.

Author Rebuttals to First Revision:

Referee #1 (Remarks to the Author):

The authors have addressed the points I raised in my review of their first draft. I recommend that this revision be published. Congratulations!

Referee #2 (Remarks to the Author):

The revised version satisfactorily addressed my concerns. I (and the potential readers as well, I believe) appreciate the authors' careful and comprehensive data reduction and statistical analysis. The detection of the 2175 Angstrom extinction bump at $z=6.7$ is robust and will be of big impacts in a range of scientific topics. I am happy to recommend the acceptance of this manuscript for rapid publication in Nature.

Regarding the relatively long central wavelength of the extinction bump (at 2263 Angstrom, instead of the canonical 2175 Angstrom), I believe, it may be related to the physical conditions. Blasberger et al. (2017, ApJ, 836, 173) reported that the interstellar extinction bump of some lines of sight toward some Herbig Ae/Be stars

peaks at substantially longer wavelength (see their Table 1). The exact reason remains unknown, probably related to different mixtures of PAHs (or ultrasmall aromatic grains like graphitic nanoparticles). So, the fact that at $z=6.7$ the central wavelength of the 2175 Angstrom extinction bump peaks at a longer wavelength actually is itself interesting and will stimulate interest in laboratory astrophysics studies of cosmic dust analogues (like carbon onions, PAHs etc, see Li et al. 2008, MNRAS, 390, L39). The authors may add a sentence or so on this. But it is also fine if the authors decide not to discuss. I am pretty sure that shortly theorists and experimentalists will explore this issue.

- We have slightly modified the text to briefly discuss the shift in wavelength could be suggestive of a different grain mixture.

There are several very minor issues:

Abstract:

``... often thought to be polycyclic aromatic hydrocarbons (PAHs) produced on times scales ..."  ``... often thought to be polycyclic aromatic hydrocarbons (PAHs) nano-sized graphitic grains produced on times scales ..."

- Having removed this sentence from the abstract as suggested, we instead modified the text as suggested by the referee (noting from the referee's point above that it should be PAHs *or* nano-sized graphitic nanoparticles) where PAHs are first mentioned (the point below).

Main text, 2nd paragraph, ``... and is attributed to carbonaceous dust grains, specifically PAHs or graphite^{1,16}." Reference #16 is not very relevant. I suggest to refer to #9 and #15, instead of #1 and #16. Actually, #16 could be deleted.

- We have changed the references as suggested.

Main text, 3rd paragraph, ``... furthermore suggest JADES-GS-z6-0 has undergone ...", furthermore  further

References:

#10 (Cox et al.) not very relevant, could be deleted.

#27 Draine, B.T. 1989, not ``Draine, Allamandola, Tielens (eds)". This was a review article by Draine alone, published in an IAU Symp. Proceedings edited by Allamandola & Tielens

There are some math symbols did not show up in the references, e.g., #8, ..of $z > 7$ Lyman break galaxies

#13, ... at redshifts $z > 10$...

#18, ... at $z > 2$...

#28, ... at $0.1 < z < 3$

...

I believe that these were caused by latex and could be easily fixed by adding $\$. \$$.

- These issues have all been fixed.

Referee #3 (Remarks to the Author):

I thank the authors for responding so completely to my previous feedback and concerns. I was particularly impressed by the following point the authors make in their updated methods section:

"Finally, we have verified that we are able to perform a blind selection from the parent sample of sources with the highest Balmer decrements and reddest UV slopes which results in a tentative detection of the UV bump. Specifically, requiring a Balmer decrement of $H\alpha/H\beta \geq 4$ and UV slope of $\beta_{UV} \geq -2.2$ yields a sample of four sources all contained within the bump sample (i.e. JADES-GS+53.16871-27.81516,

JADES-GS+53.13284-27.80186, JADES-GS+53.17022-27.77739, and JADES-GS+53.16743-27.77548; see Table 1) but notably excludes JADES-GS-z6-0."

I only have a few minor questions and suggestions, which I list below:

1) *The measured H α /H β ratio for JADES-GS+53.13423-27.76891 at $z=7.049$ is 2.11 ± 0.14 , 5 sigma below the intrinsic ratio 2.86. Could the Balmer decrement measurement be impacted by the fact that the H α line lies at ~ 5.3 microns close to where the sensitivity of NIRSpec cuts off? Is this worth remarking on?*

- Upon further inspection, we found that at this redshift the H α line is indeed only partially observed in the NIRSpec data. We remark on this seemingly unphysical Balmer decrement in the text and the table legend.

2) *While I appreciated the authors providing a brief discussion of the available far-IR ALMA constraints for sources, I think it would improve the paper slightly if the current text in the Methods section could be made a bit more quantitative. Are ALMA observations available for every source in their "bump" sample -- or are there some sources with no coverage? Is ALMA coverage available from only one program (ASAGAO) or also from other programs, e.g., ALMA GOODS (PI: Elbaz) or ASPECS (PI: Walter)? Also could the authors list the data set-dependent continuum sensitivities available to probe the dust-continuum emission from sources in their "bump" sample?*

- We agree with the referee that listing a more detailed overview of available ALMA observations and their sensitivities would benefit the discussion in this section. In our analysis, we made use of the combined data of the ALMA HUDF (PI: F. Walter), GOODS-ALMA and ASAGAO surveys (presented by Hatsukade et al. 2018). We additionally considered the ASPECS survey, which however only covers a small subset of the sources in our sample. This is now explained in the text.

Outside of the two minor issues listed above, I find the presentation to be reasonably convincing overall and so am happy to recommend to Nature to publish the result.